**Increase in flood risk resulting from climate change in a developed urban watershed–**
**The role of storm temporal patterns**
Suresh Hettiarachchi[1], Conrad Wasko[1], and Ashish Sharma[1*]
[1]School of Civil and Environmental Engineering, University of New South Wales, Sydney,
Australia.
*Corresponding author, Prof. Ashish Sharma, A.Sharma@unsw.edu.au
**Abstract**
Effects of climate change are causing more frequent extreme rainfall events and an increased
risk of flooding in developed areas.  Quantifying this increased risk is of critical importance
for the protection of life and property as well as for infrastructure planning and design. The
updated NOAA Atlas 14 intensity-duration-frequency (IDF) relationships and temporal
patterns are widely used in hydrologic and hydraulic modelling for design and planning in the
USA. Current literature shows that a rising temperatures as a result of climate change will
result in an intensification of rainfall.  These impacts are not explicitly included in the NOAA
temporal patterns , which can have consequences on the design and planning of adaptation
and flood mitigation measures. In addition there is a lack of detailed hydraulics modelling
when assessing climate change impacts on flooding.  The study presented in this manuscript
uses a comprehensive hydrologic and hydraulic model of a fully developed urban/suburban
catchment to explore two primary questions related to climate change impacts on flood risk:
(1) How do climate change effects on storm temporal patterns and rainfall volumes impact
flooding in a developed complex watershed?  (2) Is the storm temporal pattern as critical as
the total volume of rainfall when evaluating urban flood risk?  We use the NOAA Atlas 14
temporal patterns along with the expected increase in temperature for the RCP8.5 scenario for
2081-2100, to project temporal patterns and rainfall volumes to reflect future climatic change.
The model results show that different rainfall patterns cause variability in flood depths during
a storm event. The changes in the projected temporal patterns alone increase the risk of flood
magnitude upto 35 % with the cumulative impacts of temperature rise on temporal pattern
and the storm volume increasing flood risk from 10 to 170 %.  The results also show that
regional storage facilities are sensitive to rainfall patterns that are loaded at the latter part of
the storm duration while extremely intense short duration storms will cause flooding at all
locations. This study shows that changes in temporal patterns will have a significant impact
on urban/suburban flooding and need to be carefully considered and adjusted to account for
climate change when used for design and planning future stormwater systems.

## 1    Introduction

Recent history shows that extreme weather events are occurring more frequently and in areas
that have not had such events in the past (Hartmann et al., 2013). There are more land regions
where the number of heavy rainfall events has increased compared to where they have
decreased (Alexander et al., 2006; Donat et al., 2013; Westra et al., 2013a). Intensification of
rainfall extremes (Lenderink and van Meijgaard, 2008; Wasko and Sharma, 2015; Wasko et
al., 2016b) and their increasing volume (Mishra et al., 2012; Trenberth, 2011) has been linked
to the higher temperatures expected with climate change. This increase in the likelihood of
extreme rainfall and its intensification creates a higher risk of damaging flood events that
cause a threat to both life and the built environment, particular in urban regions where the
existing infrastructure has not been designed to cope with these increases. Adapting to future
extreme storm events (i.e. flood events) will be costly both economically and socially (Doocy
et al., 2013).  Properly addressing this increased flood risk is all the more important given the
expectation  that the urban  population is projected to grow from the current 54 % to 66 % of
the global population by the year 2050  (United Nations, 2014).
Adaptation as a way to address the effects of climate change has only recently gained
attention (Mamo, 2015).  Adaptation in the context of flood risk involves taking practical and
proactive action to adjust or modify stormwater management infrastructure such as low
impact development (LID) methods to reduce surface runoff or constructed storages to handle
the increased flows during an extreme storm.  The foundation of adaptation measures to deal
with flooding is typically based on flood forecasting and hydrologic/hydraulic (H/H)
modelling (Thodsen, 2007). The effectiveness of adaptation is dependent on the accuracy of
simulating projected impacts, such as the effectiveness of a flood control structure to protect
a city from future increased flooding. In addition, variability and uncertainty related to these
flood forecasts play an important role since uncertainty in future projection limits the amount
of adaptation that society will accept (Adger et al., 2009). Prior to the advent of computers
and the increase in computational power, drainage design was based on simple empirical
models of peak discharge rates using methods such as the rational formula in combination
with Intensity-Duration-Frequency (IDF) curves (Adams and Howard, 1986; Nguyen et al.,
2010). Consideration of the environmental impacts related to flow rates, volumes, water
quality and downstream impacts requires more complex systems and ways to simulate the
hydrologic and hydraulic processes in a more realistic manner (Nguyen et al., 2010).  As
such, the state of the art in modelling urban sewer and stormwater related infrastructure uses
distributed, fully dynamic, hydrologic and hydraulics modelling software (Singh and
Woolhiser, 2002).  The dynamic approach and integrated nature of current modelling requires
the use of temporal patterns to distribute rainfall and volumes that closely resemble actual
storm events (Nguyen et al., 2010; Rivard, 1996).
Temporal patterns have typically been derived using the alternating block method from IDF
curves where shorter storm durations are nested within longer storm duration design
intensities (García-Bartual and Andrés-Doménech, 2016; Victor Mockus and E. Woodward,
2015) . However, this method does not represent a real storm structure. Alternatively, Huff
(1967) presented the first rigorous analysis of rainfall temporal patterns (García-Bartual and
Andrés-Doménech, 2016), where rainfall temporal patterns were derived from observations.
Similar methods include the average variability method, where a storm is partitioned into
fractions of equal time, and each fraction is ranked. The temporal distribution is then
specified as the most likely rainfall order with the average rainfall used for the associated
fraction (Pilgrim, 1997).  NOAA Atlas 14 provides an updated set of temporal distributions
and IDF curves for use in a major portion of the United States (Perica, 2013) that are now
widely used for planning and design modelling analysis.  These temporal distributions and
rainfall depths are based on observed data and were generated using methodology similar to
Huff (1967).  The major concern is that the analysis and methods used in Atlas 14 assumes a
stationary climate over the period of observation and application (Chapter 4.5.4 of Atlas 14
volume 8).  This seems contrary to prevailing scientific thought (Milly et al., 2007) and can
lead to inadequacies of future stormwater infrastructure as there is evidence to believe that
warmer temperatures are forcing intensification of temporal patterns  (Wasko and Sharma,
2015) and an increase in variability (Mamo, 2015).  Several previous studies have examined
the sensitivity of urban catchments to changes in intensity and temporal patterns with peak
runoff rates and volumes modelled (Lambourne and Stephenson, 1987; Mamo, 2015; Nguyen
et al., 2010; Zhou et al., 2016).  For example, Lambourne and Stephenson (1987) presented a
comparative model study to look at the impact of temporal patterns on peak discharge rates
and volumes. However, with the exception Zhou et al. (2016), these studies largely ignored
the detailed hydraulic conveyance aspects of storage ponds, sewers, culverts, and flow
control structures which play an important role in how the flow rates generated during runoff
move through and impact on the built environment.
Although there are an increasing number of catchment/basin scale and urban modelling
studies that have been performed (Cameron, 2006; Graham et al., 2007; Leander et al., 2008;
Zhou et al., 2016; Zope et al., 2016), there remains a lack of a detailed studies that looks at
assessing future flood damage in a developed environment (Seneviratne et al., 2012). The
majority of past studies focus on either the hydrologic modelling component or the rainfall
intensity aspect and mostly overlook the crucial detail of rainfall patterns. In this study, we
focus on the range of results generated from detailed H/H modelling arising from
precipitation pattern variability and the impact of climatic change. We pay particular
attention on assessing and illustrating the variability in how different catchments respond to
different rainfall patterns and the impacts of climate change.  The primary questions that we
address are;
1. What is the relative importance of the storm pattern and volume of rainfall on urban
flood peaks?
2. How will climate change affect storm patterns and volumes and what are the impacts
on urban flood peaks?

Flood risk assessment and communication depend on flood risk mapping, for which flood
inundation areas are needed (Merz et al., 2010). Urban catchments are typically complex and
need to capture the response of the system along with the interactions of the various
components of the stormwater infrastructure (Zoppou, 2001) to provide reliable flood depths
to develop inundation areas. The main characteristic of stormwater in urban areas is that the
flows are predominantly conveyed in constructed systems, replacing or modifying the natural
flow paths.  Including the complex hydraulics and possible hydraulic attenuation and timing
of congruent flows will have an impact on flooding, particularly in developed environments.
As discussed, temporal patterns of rainfall is now a critical aspect of design and planning of
future storm systems.  Research which uses temperature to project future rainfall and
temporal patterns, and then assesses impacts on flooding has not been performed.  This study
aims to fill this research gap through an elaborate analysis of how rainfall intensities and
patterns impact urban flood risk in a warmer climate.

## 2.    Assessing flooding in developed/urban stormwater systems

Developed urban areas present the highest probability of causing damage and loss of life during flood events. There has been an increase in urban flooding in the past decade with densely populated developing countries like India and China coming into focus (Bisht et al., 2016; Zhou et al., 2017). A case study on the Oshiwara River in Mumbai, India has shown a 22 % increase in the overall flood hazard area due to changes in land use and increased urbanization within the catchment (Zope et al., 2016). In particular, flooding in Mumbai in 2005, which was caused by extreme rainfall coupled with inadequate storm sewer design, is blamed for 400 deaths (Bisht et al., 2016). China has also experienced a devastating flood season in 2016 (Zhou et al., 2016) with the rapid increase in urbanization. Even with better planned and mature urban cities, Europe and North America are not immune to flooding in urban areas (Ashley et al., 2005; Feyen et al., 2009; Smith et al., 2016). Impacts of climate change are expected to increase the risk of flooding and further exacerbate the difficulty of flood management in developed environments.

## 3    Assessing climate change impacts on flooding

The number of studies investigating climate change impacts on urban flooding is increasing as the importance of this topic is more and more recognized. However, research focusing on the impacts of climate change on precipitation temporal patterns remains limited. The majority of available research use Global Circulation Models (GCMs) and Regional Climate Models (RCMs) combined with statistical downscaling techniques to project IDF curves to reflect future climate conditions (Mamo, 2015; Nguyen et al., 2010; Schreider et al., 2000). For example Mamo (2015) used monthly mean wet weather scenario data projected by four GCMs for the period 2020-2055, along with historic data from 1985 to 2013, which were then used as weather generator input using LAR-WG, from which data was generated to develop revised IDF curves. Nguyen et al. (2010) used data sets generated by two separate GCMs to develop IDF and temporal patterns to reflect future rainfall patterns. The inconsistent results generated by the two different GCMs illustrate the challenge of forecasting future climate conditions with GCM generated results. It is recognized that GCM results form the largest part of the uncertainty in projected flood scenarios (Prudhomme and Davies, 2009).

Alternatively, research has shown that temperature, which influences the amount of water contained in the atmosphere, can have an impact on the patterns and total rainfall volumes of

storm events (Hardwick Jones et al., 2010b; Lenderink and van Meijgaard, 2008; Molnar et
al., 2015; Utsumi et al., 2011; Wasko et al., 2015; Westra et al., 2013a). In general,
intensification of rainfall events is expected with a trend towards 'invigorating storm
dynamics" (Trenberth, 2011; Wasko and Sharma, 2015). Even though forecasts for climate
change impacts on future flooding have a 'low confidence', global scale trends in temperature
extremes are more reliable (Seneviratne et al., 2012). Following successful studies (Wasko
and Sharma, 2017; Westra et al., 2013b) we take the approach of using temperature to project
temporal patterns and rainfall volume to account for climate change impacts. As described in
detail in section 5, we examine historical rainfall data coupled with daily average temperature
to project temporal patterns and rainfall volumes to account for climate change impacts.
**4.     Study location, data and methodology**
In this study we use temperature to project rainfall temporal patterns and volumes to evaluate
the variability in flood risk as well as the impact to flood risk due to climatic change.
Broadly, the steps followed are:

1   Apply multiple temporal patterns and rainfall volumes with their associated

confidence limits in the H/H model to establish the variability in the flood risk

2   Develop scaling factors (Lenderink and Attema, 2015; Wasko and Sharma, 2015) for

the volume and temporal pattern for future conditions using temperature as index

3   Evaluate the impact of temperature rise on flood risk by scaling temporal patterns for

a temperature increase

4   Evaluate the cumulative impact of temperature rise on flood risk by scaling both

volume and temporal patterns

The hydrologic and hydraulic modelling performed here used the EPA-SWMM model of an
urban/suburban catchment in Minneapolis, Minnesota, USA. The SWMM software package
was initially developed by the United States Environmental Protection Agency (EPA, 2016)
and has since been used as the base engine for most of the industry standard H/H modelling
packages.
**4.1    Study Location and model**
The H/H model used in this study was developed for the South Washington Watershed
District (SWWD) in the State of Minnesota, USA for the management of surface water flows
and as well as for planning and management of on-going development work and capital
improvement projects. The catchment area of the SWWD is a highly developed
urban/suburban area and extends over 140-km$^2$. The model was initially built in the year
2000 and has been continuously maintained and updated with the latest available
landuse/land cover and stormwater infrastructure information. The model includes extensive
detail of all landuse types and stormwater infrastructures including sewers, culvert crossing,
open channel reaches, and constructed as well as natural storages. Highly detailed
delineation of both sub-catchment boundaries and impervious area was done using a high
resolution Digital Elevation Model (DEM), development construction and grading plan
overlays and aerial imagery within a GIS environment. All surface runoff is fed into the
appropriate inflow points of the hydraulic conveyance system. The model has been validated
and used to design major capital improvement and flood mitigation projects (Hettiarachchi et.
al. 2005, Hettiarachchi and Johnson, 2006). Additional model information is available in the
supplemental information section S1. For the purposes of this study and to reduce the
complexity and model run times, the model was trimmed to the upper section of the SWWD
representing an area of approximately 22 km2.
Figure 1 presents the focus areas along with the schematic of the model network to illustrate
the level of detail of the existing storm water infrastructure captured in the model. As
discussed above the model includes geometry details to explicitly model the street overflow
routes where flooding occurs as well as depth/area curves that capture flooding at the storage
nodes. This level of detail results in accurately modelling the travel time of flows within the
watershed and capturing all the runoff volume generated from the storm. Additionally, the
geometry detail provides a reasonably accurate representation of extents related to flooding.
The proper simulation of hydraulic attenuation and a variety of landuse types provide an ideal
platform for this study.
Table 1 lists the primary reference locations that are used for this study. The locations have
specifically been chosen to represent the range of possible conditions that are encountered in
urban catchments. The sub-catchment sizes vary from less than 0.5 km$^2$ to approximately
2 km$^2$, with an overall catchment of 22 km$^2$. Different land uses such as commercial and
industrial or different types of residential areas, as well as the amount of storage, have all
been considered. It is important to note that these locations were selected prior to any model
runs or availability of results and hence do not bias the results presented. Table 1 gives a
description of the primary landuse type of the subwatershed that drains to each reference
location along with the watershed area and the overall percentage of impervious surface area
within that watershed.  It also describes if there are local storage ponds, either natural or
constructed, that provide rate and volume control.

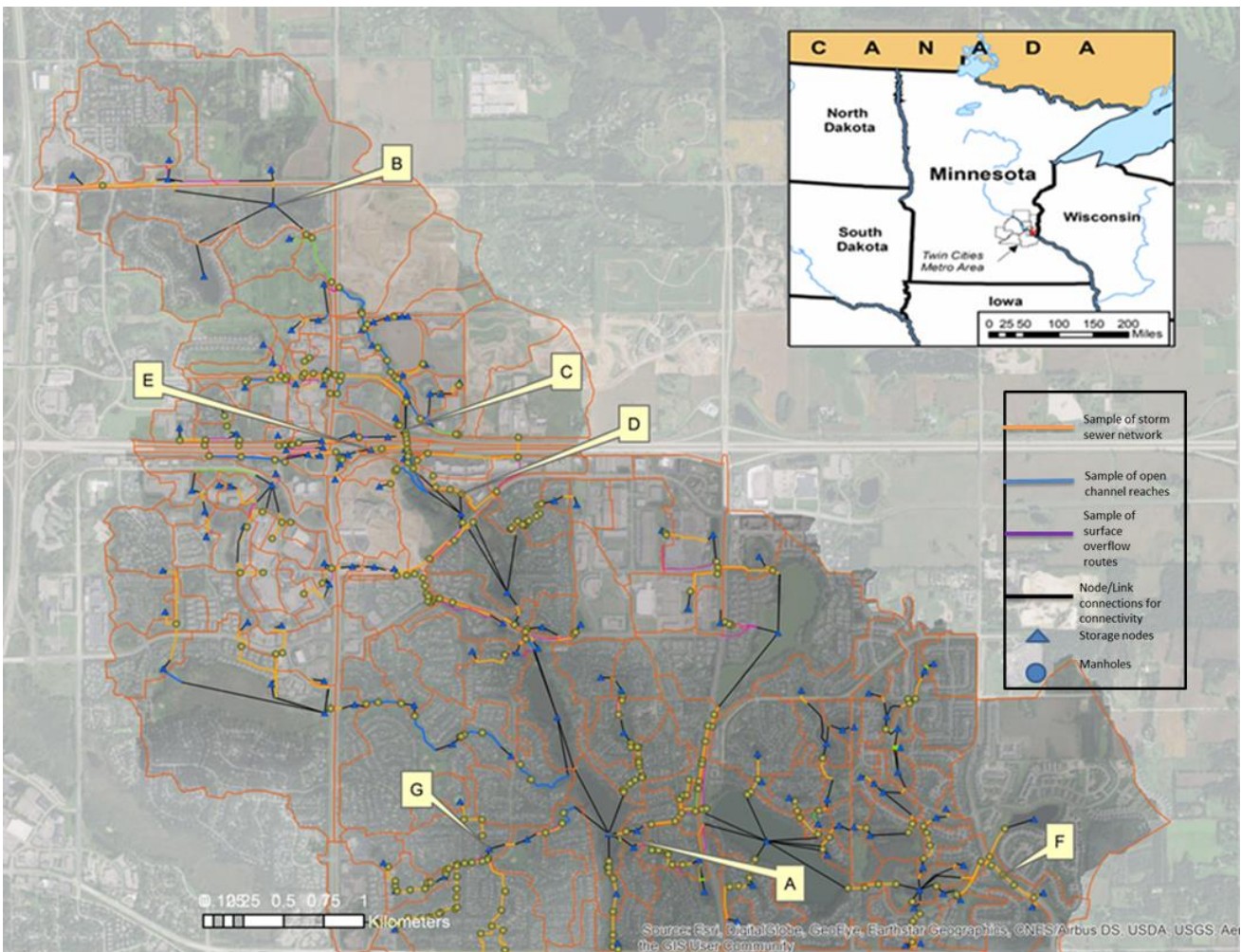


**Figure 1 Location of the model and the sub-watersheds along with the reference points used in the**
**discussion below.  The details of the reference points and further explanation are presented in Table 1.**
**The Orange links are example of the sewer network geometry in the model.  The blue links represent**
**reaches that are open channel.  The magenta links are the surface overflow routes that capture flow that**
**tends to flood in areas and spread outside the sewer network.  The black links provides connectivity when**
**the georeferenced locations of nodes are geographically different to the ends of some of the sewer**
**network.  The black links provide connectivity in the model.**




**Table 1. Description of reference locations presented in Figure 1 and used to present results. Each**
**location represents a variation of landuse within the watershed**

| Reference point | Landuse types and description | Watershed Area (km$^2$) | Average Percent impervious |
|---|---|---|---|
| (A) Wilmes | Natural lake and downstream limit of watershed. | ~ 22 | - |
| (B) Upstream | Predominantly rural, lower density residential landuse with good tree canopy and green spaces. Natural wetlands to mitigate flow with minimal to constructed storage | 2.2 | 32 |
| (C) Business park | Office space and parking lots with green space mixed in. Constructed storage and infiltration to help mitigate runoff | 0.5 | 42 |
| (D) Commercial 1 | Retail and parking dominates this area with some green spaces added in. Minimal constructed storage. Two sub-surface infiltration basins installed under parking lots | .25 | 60 |
| (E) Commercial 2 | Retail and parking dominates this area with substantial constructed storage to help mitigate runoff rates and volumes. Part of the highway also drains through this point. | .75 | 48 |
| (F) Residential 1 | Medium density residential landuse with minimal constructed storage. | .35 | 24 |
| (G) Residential 2 | Medium density residential landuse with constructed storage. | 1.05 | 39 |


## 4.2 Precipitation and Temperature Data

The precipitation and temperature data used in the analysis were sourced from the National
Centers for Environmental Information hosted by the National Oceanic and Atmospheric
Administration (NOAA). Both hourly and daily rainfall data were downloaded from the
climate data online site for Minneapolis and St Paul (MSP) International Airport gauge,
which is the closest major airport to the study area. Daily data for the MSP airport was
available from 1901 through 2014, while hourly data was available from 1948 through 2014.
Daily maximum, minimum and average temperature data was also downloaded for the period
from 1901 through 2014. For this analysis days that did not have precipitation data were
assumed to have no rain.
The temporal patterns for storms and the depths of rainfall were taken from NOAA ATLAS
14 volume 8 (Perica. et. al. 2013) – the current state of the art design standard for this
location.  The modelling analysis  centred on the 50-year (2% exceedance probability) storm,
which is a total rainfall volume of 160 mm in 24-hours, for the area within the SWWD in the
USA.  The 90 % confidence margin storm depths were added to the analysis to look at how
modelled flood depths vary with total precipitation (Table 2).  Six temporal distributions (two
patterns with their associated confidence margins) were chosen from NOAA ATLAS 14
volume 8 to investigate how flood depths are impacted by the shape of storm over a 24-hour
period.  Table 2 describes the different storm temporal patterns and each of the precipitation
volumes modelled.  The spatial distribution of rainfall was kept constant for this study.  Even
though we acknowledge that spatial variability of rainfall will have a significant impact on
flooding, adding that dimension to the current analysis would have made the level of effort
excessive.  Also, by not spatially varying the rainfall distribution, we are able to better focus
on the sensitivity of temporal patterns on flooding impacts.










**275 Table 2. Description of notation used in reference to the modelled storm depths and**

**276 temporal distributions (NOAA Atlas 14 volume 8 appendix 5)**

| Design Rainfall | Description |
| --- | --- |
| 160 mm  24 hour | 2 % exceedance 24-hour duration (50-year return period) rainfall depth |
| 125 mm 24 hour | Lower margin of the 90% confidence interval of the 2 % exceedance 24-hour duration (50-year return period) rainfall depth-Approximately Equivalent to the 20-year 24 hour ARI |
| 210 mm 24 hour | Upper margin of the 90% confidence interval of the 2 % exceedance 24-hour duration (50-year return period) rainfall depth-Approximately Equivalent to the 200-year 24 hour ARI |
| **Temporal pattern** | **Description** |
| Q1-10 - (a) | NOAA Midwest region, $1^{st}$ quartile $10^{th}$ percentile temporal distribution |
| Q1-50 - (b) | NOAA Midwest region, $1^{st}$ quartile $50^{th}$ percentile temporal distribution |
| Q1-90 - (c) | NOAA Midwest region, $1^{st}$ quartile $90^{th}$ percentile temporal distribution |
| Q3-10 - (d) | NOAA Midwest region, $3^{rd}$ quartile $10^{th}$ percentile temporal distribution |
| Q3-50 - (e) | NOAA Midwest region, $3^{rd}$ quartile $50^{th}$ percentile temporal distribution |
| Q3-90 - (f) | NOAA Midwest region, $3^{rd}$ quartile $90^{th}$ percentile temporal distribution |


The quartiles indicate the timing of the greatest percentage of total rainfall that occurs during
a storm.  First quartile indicates that the majority of the rainfall, including the peak, occurs in
the 1st ¼ of the duration, which is between hours 1 through 6 in the case of a 24-hour storm.
Third quartile indicates that the majority of the rainfall, including the peak occurs in the $3^{rd}$
quarter of the storm duration, that is, hours 12 through 18 in the case of a 24-hour storm. The
temporal distributions were also separated in Atlas 14 to determine the frequency of
occurrence within each quartile to determine a percentile for each distribution.

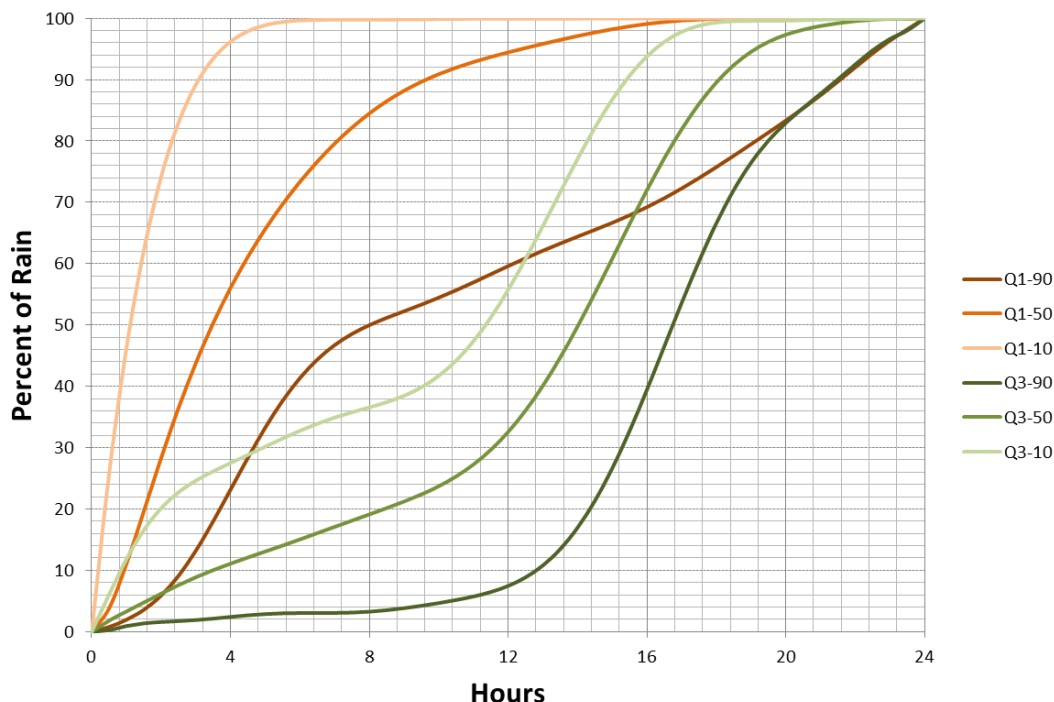


**Figure 2. NOAA Atlas 14 temporal patterns used in the modelling**


The SWMM model was run for each of the precipitation amounts for the six temporal
patterns, a total of 18 model runs, to generate the base dataset for current conditions and
establish the variability in the current climate.  The impact of climate change due to changed
temporal patterns was assessed by modelling the 2% exceedance rainfall value (160 mm)
with temporal patterns scaled for an expected temperature increase. Finally the cumulative
impacts of changed temporal patterns and volume were evaluated by scaling both the rainfall
volume and temporal patterns with temperature.  An important point to note is that only the
rainfall time series was changed appropriately for each model run. All the boundary
conditions such as initials water levels at storage locations and all hydrologic parameters for
each of the above model runs were kept the same for every model run.

### 4.3    Temperature scaling of temporal patterns and rainfall volume

To assess the impact of climate change, design storm temporal patterns and rainfall volumes
need to be projected for a future warmer climate. Most methods that project rainfall for future
climates focus on downscaling output from general circulation models to those required for

hydrological applications (Fowler et al., 2007; Maraun et al., 2010; Prudhomme et al., 2002)
through either dynamical or statistical models (Wilks, 2010). Downscaling methods,
however, will not replicate design rainfall (Woldemeskel et al., 2016), so an attractive
alternative is that proposed by Lenderink and Attema (2015) whereby historical temperature
sensitivities (scaling) are directly applied to the design rainfall. Here, we assume that
temperature is the primary climatic variable associated with changing rainfall. This is
consistent with studies that find that temperature is a recommended covariate for projecting
rainfall (Agilan and Umamahesh, 2017; Ali and Mishra, 2017) and temperature sensitivities
implicitly account for dynamic factors (Wasko and Sharma, 2017). Indeed projecting rainfall
directly using temperature sensitivities gives comparable results to more sophisticated
methods of rainfall projection using numerical weather prediction (Manola et al., 2017).
Using established methods (Hardwick Jones et al., 2010a; Utsumi et al., 2011; Wasko and
Sharma, 2014), the volume scaling for the 24 hour storm duration was calculated using an
exponential regression. The results are presented in Figure 5. First, daily rainfall was paired
with daily average temperature. The rainfall-temperature pairs were binned on 2℃
temperature bins, overlapping with steps of one degree. For each 2℃ bin a Generalized
Pareto Distribution fitted to the rainfall data in the bin that was above the 99th percentile to
find extreme rainfall percentiles (Lenderink et al., 2011; Lenderink and van Meijgaard,
2008). Extreme percentiles below the $99^{th}$ percentile (inclusive) were calculated empirically.
A linear regression was subsequently fitted to the fitted log-transformed extreme percentiles
and used as the rainfall volume scaling (Figure 5). Hence the volume (V) is related to a
change in temperature (T) by

$$V_2 = V_1(1 + \alpha)^{\Delta T}$$


Where α is the scaling of the precipitation per degree change in temperature.
Temporal pattern scaling was calculated using hourly data, again paired to the average daily
temperature and followed the proposed methodologies in (Wasko and Sharma, 2015). The
largest 500 storm bursts of duration 24 hours were identified in the hourly data, with each
storm burst independent (not overlapping). The 24-hour duration storm bursts were divided
into 6 fractions, each fraction with duration of four hours. Each fraction was divided by the
rainfall volume and ranked from largest to smallest. An exponential regression was fitted to
the fractions corresponding to each rank and their corresponding temperature to produce a
temporal pattern scaling. The scaled temporal patterns were then applied and run through the
H/H models.
**5**     **Results and Discussion**
The results from the modelling analysis is presented and discussed below. We show that the
current temporal patterns for design flood estimation need to be adjusted to account for
climate change impacts as do design rainfall volumes.
**5.1**     **Temporal patterns and volume scaling**
The scaling of the temporal pattern fraction for Minneapolis is presented in Figure 3. Table 3
provides the scaling that results from the fitted regression in each of the panels in Figure 3. A
temperature change of $5°C$ was selected to determine the percentage change based on
temperature increases estimated for the RCP8.5 scenario in Figure SPM7(a)(IPCC 2014)
projected for 2081-2100. The selection of the RCP8.5 scenario was based on the goal of this
paper to demonstrate the importance of accounting for climate change in rainfall patterns as
well the current literature suggesting that we are tracking on a RCP8.5 scenario (Peter et al.,
2013). Additional analysis performed for the RCP4.5 scenario (supplemental information
section S2) shows similar trends in results but of a lesser magnitude. It is important to note
that rigorous thought is needed on how far out and what level of climate impacts should be
considered when selecting a threshold for design or when setting absolute flood depths.
As the slopes in Figure 3 and factors in Table 3 show, only the first fraction scaled positively,
which means that the 4 hours that included the highest amount of rainfall scale up while the
remaining rainfall fractions scale down. The results are consistent with "invigorating storm
dynamics" (Lenderink and van Meijgaard, 2008; Trenberth, 2011; Wasko and Sharma, 2015;
Wasko et al., 2016b) resulting in a less uniform, more intense storm. The percentage
adjustments were normalized to make sure that total rainfall amount did not change from the
current value of 160 mm in 24-hours. Figure 4 presents (Q1-50 and Q3-50 shown as an
example) the changes to the temporal patterns when the scaling percentages calculated above
are applied. Figure 4 illustrates the change to the highest peak rainfall rate and the decrease in
the rest of the rainfall fractions. Similar scaling was applied for all six temporal patterns that
were used in the H/H modelling analysis. As an additional verification, a similar analysis
was completed for two neighbouring locations (Sioux Falls South, Dakota and Milwaukee,
Wisconsin). The fraction and volume scaling results for both Sioux Falls and Milwaukee
were consistent with those discussed in this paper.
Figure 5 presents the precipitation volume temperature pairs, the extreme percentiles
generated based on the temperature bins, as well as the resulting scaling for the 24 hour
rainfalls.  The daily total rainfall of 160 mm fell into the 99.99[th] percentile based on a cursory
ranking of the daily precipitation data.  Hence, the 99.99[th] percentile 4.7 % scaling was
selecting for the 24 hour volume.  This is broadly consistent with Utsumi et al. (2011) and
Wasko et al. (2016a) who present scaling between 2 and 5 % for the central north of the U.S
for the 99[th] percentile and throughout Australia and less than the scaling found by Mishra et
al. (2012) who used hourly precipitation, which is consistent with the expectation that shorter
duration extremes have greater scaling (Hardwick Jones et al., 2010a; Panthou et al., 2014;
Wasko et al., 2015).  This value also appears to be consistent both with historical trends and
climate change projections. Barbero et al (2017) looked at a non-stationary extreme value
analysis and found a sensitivity of approximately 7%/ºC for a non-stationary Theil-Sen
estimator for North America. Globally, Westra et al. (2013) find historical trends have global
sensitivity between 5.9%/ºC and 7.7%/ºC. However, Kharin et al (2013) report an
approximately 4% sensitivity over land globally from the CMIP5 model results with a range
of 2.5-5% for the U.S.A. Relative to the literature stated above we believe our projections are
consistent with the available evidence regarding precipitation change.
This 4.7 % scaling converts to an approximately 20 % increase in the volume of rainfall in a
24 hour period for a five degree increase. Applying the 20 % increase to the 160 mm in 24-
hours gives a rainfall depth of 208 mm in 24 hours.  Coincidentally, 208 mm (~210 mm) in
24 hours is the upper margin of the 90% confidence interval for the 160 mm event based on
the margin provided in NOAA Atlas 14.

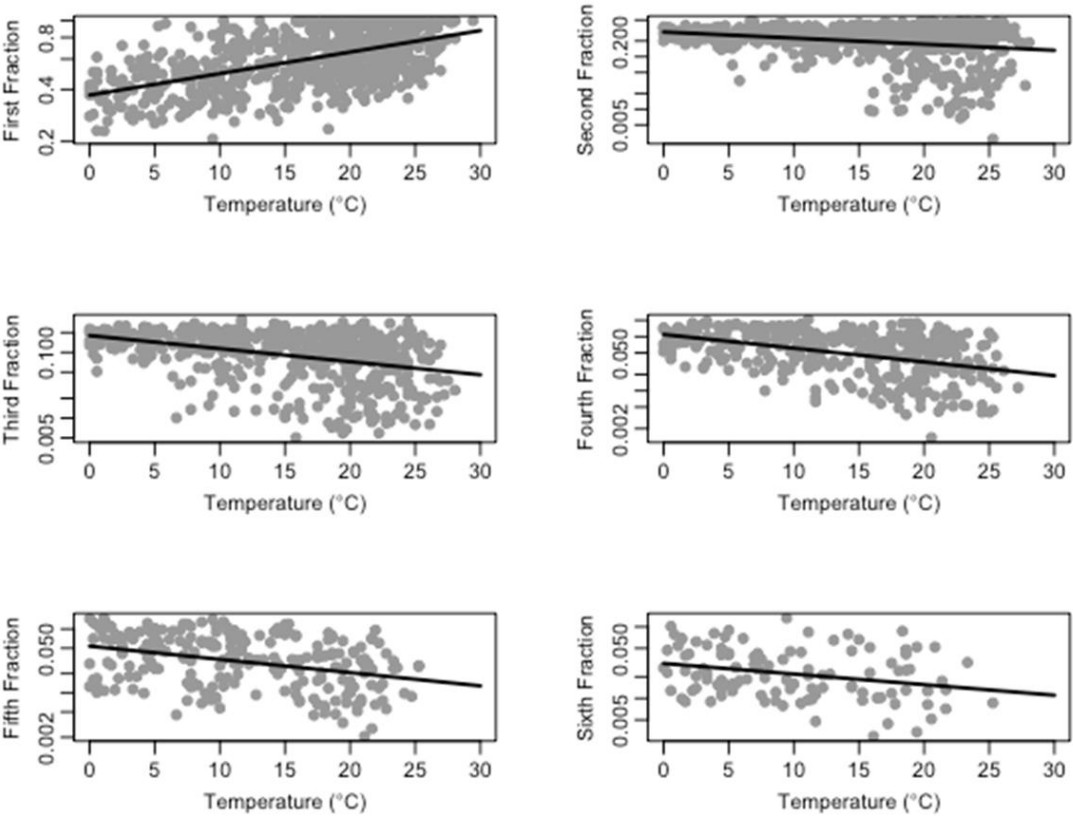


**Figure 3. Scaling temporal pattern fractions with temperature for Minneapolis (1948-2014 hourly data).**
**Black lines represent the fitted exponential regression.**
**Table 3  Temporal pattern scaling factors for each of the fractions**

| Fraction | Scaling factor |
|---|---|
| F1 | 0.029 |
| F2 | -0.026 |
| F3 | -0.045 |
| F4 | -0.057 |
| F5 | -0.047 |
| F6 | -0.033 |


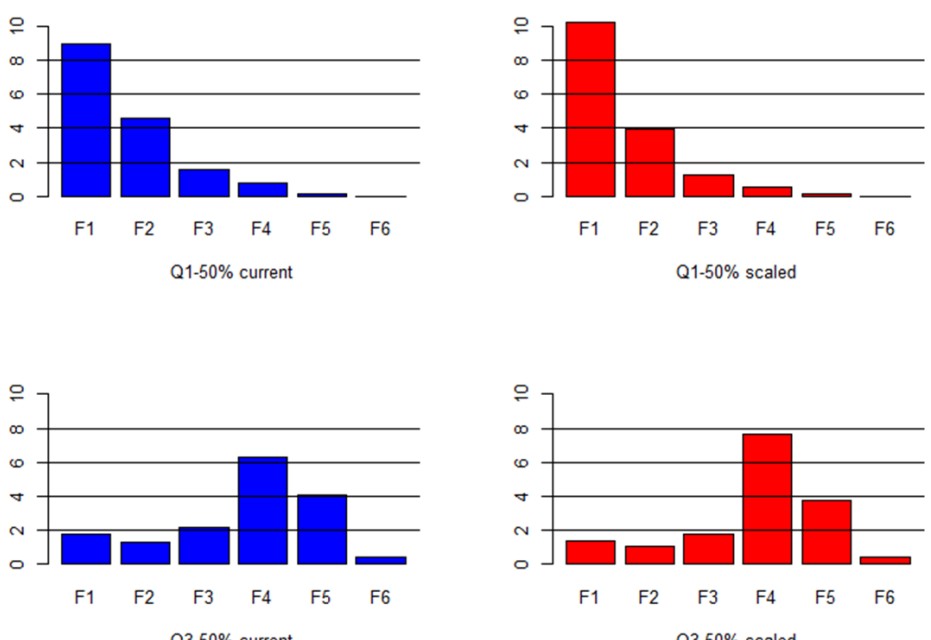


**Figure 4 Q1-50 and Q3-50 temporal patterns projected for temperature rise of $5^0$ C. Total rainfall of 160**

**mm over 24 hours with each fraction representing accumulated rain for 4-hour periods.**

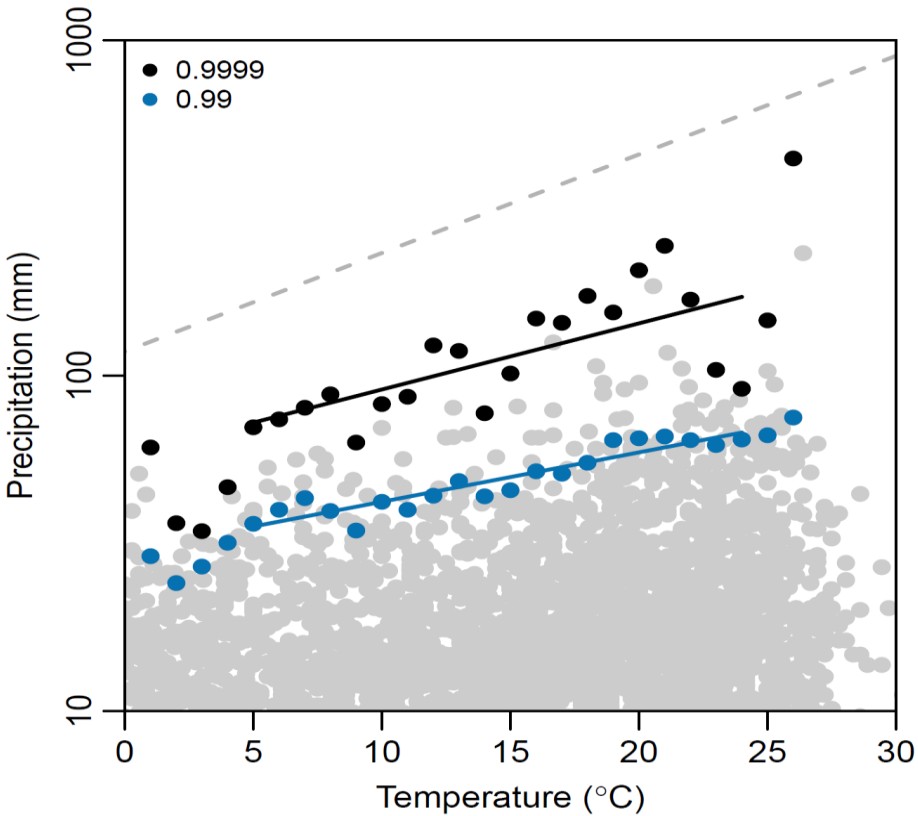

**Figure 5 Scaling total volume of rainfall with temperature for Minneapolis (1901-2014 daily rainfall).**
**Grey dots are rainfall temperature-pairs and the coloured dots are the extreme percentiles. The grey**
**dashed line represents a scaling of 7 %.**

## 5.2     Flood depth response to temporal patterns and total rainfall variability.

The hydrologic/hydraulic model was run for the 18 different combinations of rainfall
volumes and temporal patterns. Results are presented for the five reference locations
throughout the watershed representing different landuse types that are typical in a developed
area as described in Table 2.  The selection of the reference points essentially provides results
at different sub-catchments, or different sub-models.  These sub-models show the variation in
catchment response to runoff generated by a variety of land use types as well as changes in
how the flows move through the different stormwater infrastructure.

Figures 6(a) shows the depth/time curve at Wilmes Lake (location A) which is the main
regional collection point and the downstream end of the model.  Each curve represents
change in depth versus time for the six temporal patterns distributing the same total rainfall
volume of 160 mm.  The differences in shape, peak flood depth and the time to peak illustrate
the variability in catchment response that can result purely due to variation in rainfall pattern
during a storm event.  A striking result is the approximately 30 % (1.3 m) variation in flood
depth (relative to mean flood depth) at Wilmes Lake purely due to variation of how the rain
falls within the duration of the storm. The highest flood depth curve is a result of the most
intense storm event pattern which is the Q1-10 distribution.  The depth at Wilmes Lake rises
quickly during the Q1-10 event but the peak flood depth still occurs within the 40 – 60 hour
band similar to the other rainfall patterns.  The high intensity of the Q1-10 pattern can
overwhelm local conveyance and storage structures, resulting in overflows that flushes down
to the low lying areas rapidly, causing the water level at the lake to rise.  Note that the next
highest peak flood level results from the Q3 patterns which has the majority of the
precipitation loaded at the latter half of the storm event.  Comparison of the total runoff
volume for the catchment between Q1-50 to Q3-50 temporal patterns shows a 9.5% increase
for the same 50 year storm event.  A third quartile rainfall pattern can results in higher runoff
volume as the soil saturates and infiltration rates are reduced and can cause worse flooding as
local storage structures and ponds fill up by the time the bulk of the storm occurs. The results
for the Q-3 patterns suggests that regional storage facilities such as Wilmes Lake within the
SWWD are more sensitive to the runoff volume than the instantaneous peak flow rate, and
thereby more sensitive to end loaded temporal patterns during storms.
Figure 6(b) illustrates the same type of variation of peak flood depth due purely to the
different temporal patterns at all of the reference points.  Locations A, C, D and G average
about a metre in peak flood depth variation.  When considering that the typical freeboard
(added elevation above base flood elevation) used in the USA when setting lowest open
elevations for structures is 0.65 m, a 1 m variation in peak flood elevation is significant.  As
described in Table 1, the landuse within the subcatchment that drains to location B is rural
with local natural storage whereas locations C and D have commercial land use with higher
impervious land cover.  This difference in land cover can explain why the variability in peak
flood depth relative to changes in temporal patterns is lower at approximately 0.5 m and
suggests that catchments with higher impervious surfaces have a higher sensitivity to rainfall
patterns.  Additionally, locations F is within the storm sewer system which suggests that
variation in flow rates, or peak runoff from a catchment, does not always translate to higher
variation in flood depths.
The depth vs time curves in Figure 6(a) also illustrate the value of including detailed
hydraulic routing in the modelling analysis. As an example, the curves for Q1-10 and Q3-90
patterns show the difference of catchment response due to a high intensity rainfall event that
results in an initial peak flood depth resulting from overflows followed by the lagged
response of the volume accumulation compared to the scenario of higher volume of runoff
due to saturated soils.  The variability in how the catchment responds to different temporal
patterns is consistent with studies by Ball (1994) and Lambourne and Stephenson (1987).
Though these studies focused primarily on the hydrologic aspect of the modelling and peak
flow rates and volumes, the variation in catchment response to changes in "how it rains" is
similar.  The current study has the added benefit of detailed hydraulics routing and it is
reasonable to assume that using only hydrologic routing, which is more common in current
literature, would not have captured some of the detailed environmental hydraulics that can
lead to better flood estimates in developed environments.



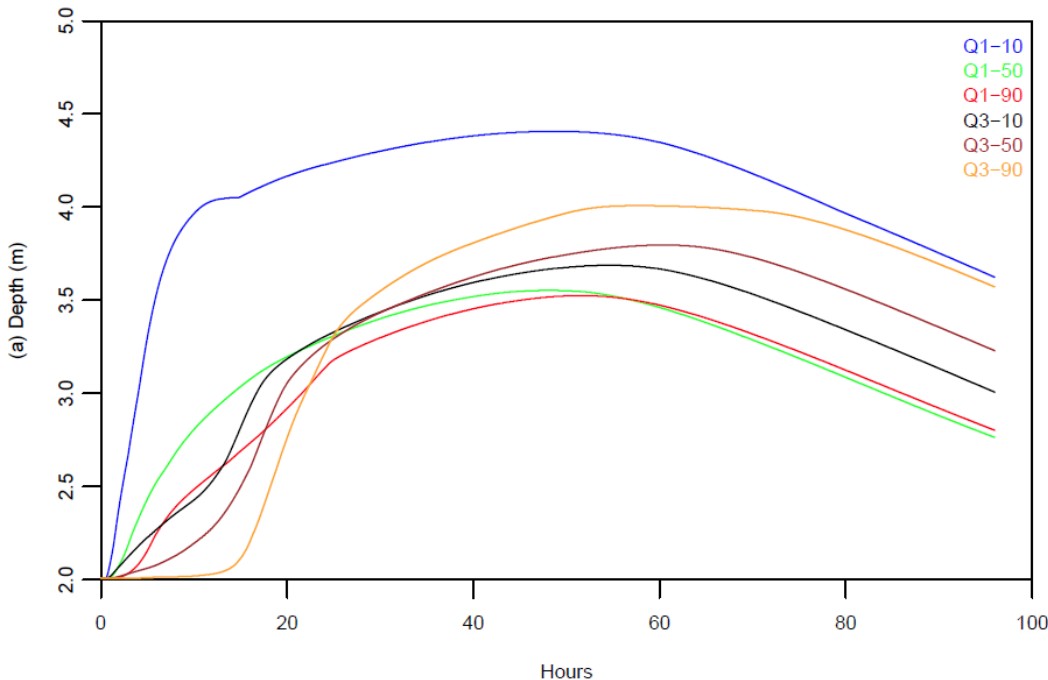


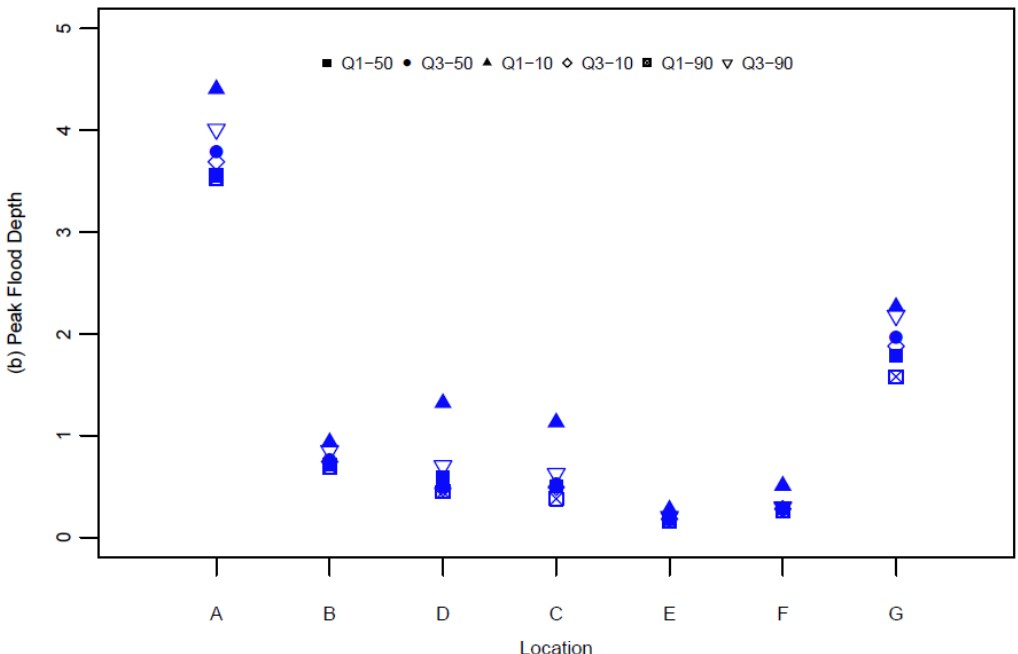


**Figure 6 (a) Depth over time at Wilmes Lake (Location A), which is the downstream regional reference point in Figure 1. Depth vs time curves are plotted for 160 mm of total rainfall over 24 hours with the six temporal patterns. (b) Presents the variation of peak flood depth (m) at reference locations throughout**


 **the watershed (ref to table 1) with variation of temporal patterns for a total of 160 mm of rainfall over 24**

**hours.**

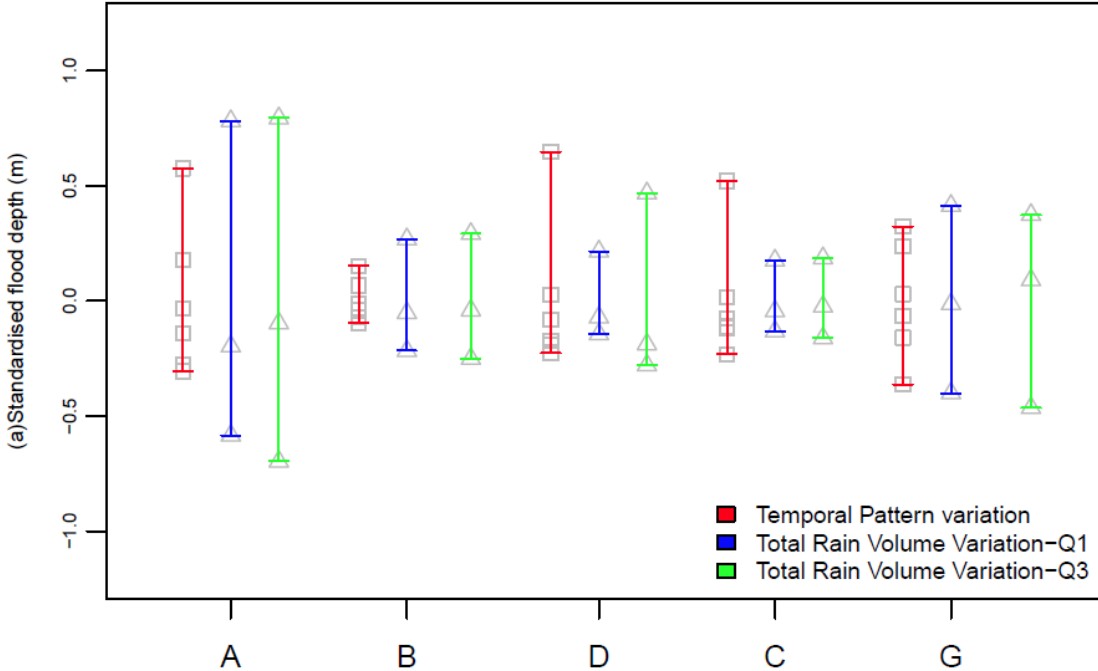


**Figure 7 Comparison of total volume of rainfall and temporal patterns variability impact on peak flood**
**depth. Flood depth variation due to the 6 different temporal patterns with 160 mm of rain compared to**
**110, 160 and 210 mm of total rainfall over 24 hours distributed over Q1-50 and Q3-50 temporal patterns.**
**Flood depths were standardised by subtracting the mean at each location for ease of comparison.**

One of the primary questions that we set out to answer was the comparison of "how it rains"
versus "how much it rains". For clarification, "how it rains" refers to the variation of
temporal patterns during a storm event with the total rainfall volume with the 24 hours held
constant. The term "how much it rains" refers to different volumes of total rainfall within 24
hours for each storm event with the temporal pattern held constant. Figure 7 makes the direct
comparison between the variations of peak flood depth between "how it rains" versus "how
much it rains". The range in peak depths at the reference locations indicates how the
different catchments respond to variability in storm volume and pattern.
Comparison of the range of peak flood depths at locations C and D indicates a higher
sensitivity to variation in "how it rains" as opposed to changes "how much it rains".
Conversely, locations A, B and G indicate a higher range in flood depths due to changes in
total rainfall volume, or "how much it rains" compared to changes in temporal patterns, or
"how it rains". Even though one can note that locations C and D receive runoff from
catchments that have a majority of higher impervious landuse relative to other locations, the
number of data points does not allow for a statistically significant comparison of the
sensitivity of impervious percentages in landuse to the difference in "how it rains" vs. "how
much it rains". But it is important to note the consistency in the range of results across all the
locations and the fact that "how it rains" has as much of an impact in the peak flood depths as
"how much it rains". The results in Figure 7 clearly answer the first question presented in the
introduction that temporal patterns of storms are as important as the total volume of rainfall
during a storm in watershed response and flood estimation.
The results presented in Figures 6 and 7 shows that temporal patterns, or "how it rains" add a
degree of variability and has a significant contribution to the overall uncertainty in H/H
modelling results. This is especially a concern given the evidence to date that systematic
change is occurring to rainfall patterns across climate zones, making them more intense and
impactful in derived flood estimations (Wasko and Sharma, 2015). The added variability has
implications on the already complex nature of properly accounting for uncertainty in flood
forecasts or the impacts of climate change in future flooding conditions, which can in turn
have implications on how society will accept the socio-economic impacts of adaption as
previously mentioned. Hence, careful consideration of "how it rains" and changes in "how it
rains" have to be included in any H/H modelling frame work along with the current typical
practice of modelling "how much it rains".

**5.3    Impact of applying temperature scaling to temporal patterns and rainfall volume**
**on flood depths**
Figure 8 compares the results for projected temporal patterns with results from the base
simulation. Both scenarios are based on the 50 year return period event which is 160 mm
distributed over the six base and projected temporal patterns. The results shown in Figure 8
are variation of the peak flood depth around the mean of the results from the base conditions
models.  In other words, the results were standardized by subtracting the mean of the base
conditions from the results at each location.

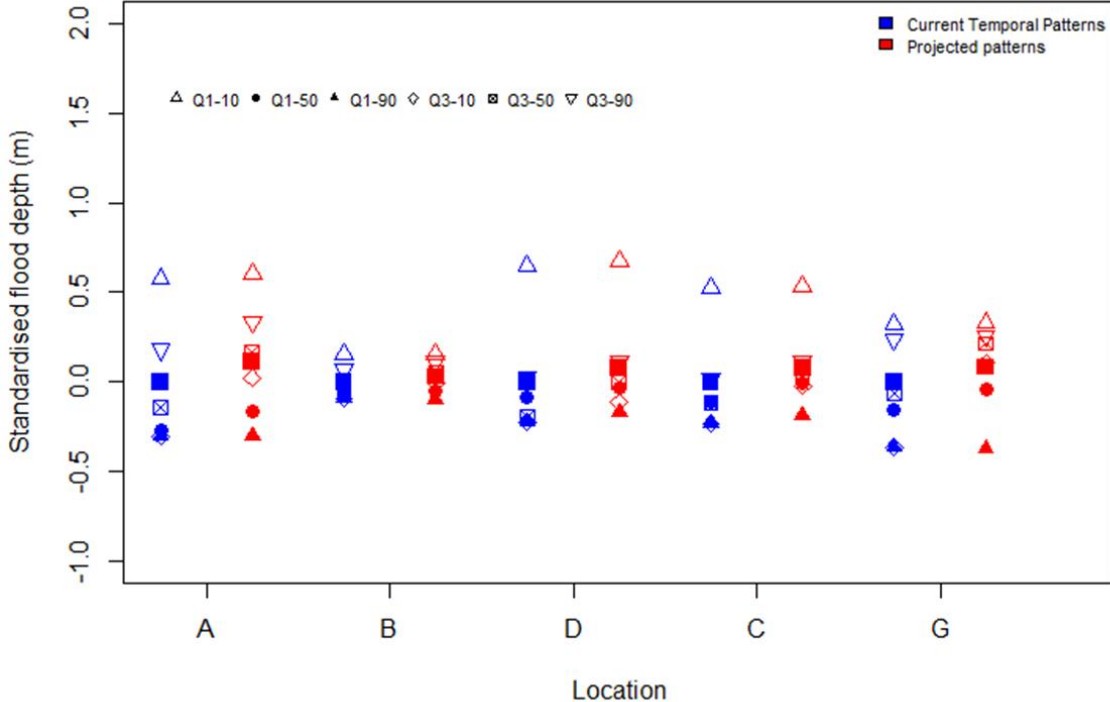


**Figure 8. Impact of rise in temperature on the peak flood depth variation at reference locations within the**
**watershed when scaling is applied only to temporal patterns.  The peak flood depths at each reference**
**point are based on 160 mm of total rain distributed over the 6 temporal patterns used.  Temperature**
**scaling (T/S) for the temporal patterns are based scaling fractions presented in Figure 3. Flood depths**
**were standardised by subtracting the mean from the base simulations presented in Figure 6 for each**
**location.**
As expected, the highest flood depth results from the Q1-10 pattern for both current and
scaled conditions. But the results at the highest depths show little change due to temperature
scaling of the Q1-10 pattern.  The Q1-10 pattern is an extremely high intensity event with
majority of the rainfall occurring in the first fraction of the event. Applying the scaling
percentages to this fraction makes minimal changes to the overall pattern of rainfall resulting
in no appreciable change in peak flood depths.  If we take the extreme Q1-10 event out of
consideration, one can say that qualitatively there is an increasing trend in flood depths due to
changes in the projected temporal patterns.  The important fact is that these plots are based on
the same total rainfall volume of 160 mm.  The moderate increasing trend in the results is
purely due to the projected temporal patterns.  As discussed previously, location B represents
a more rural type catchment and shows less sensitivity to changes in rainfall patterns.
Figure 9 shows the same comparison as in Figure 8 when temperature scaling is applied to
both the temporal pattern and rainfall volume.  Hence Figure 9 presents the cumulative
impacts of temperature scaling to the base conditions.  As in Figure 8, the results in Figure 9
show the variation of results for both scenarios around the mean of the base condition flood
depth at each location.

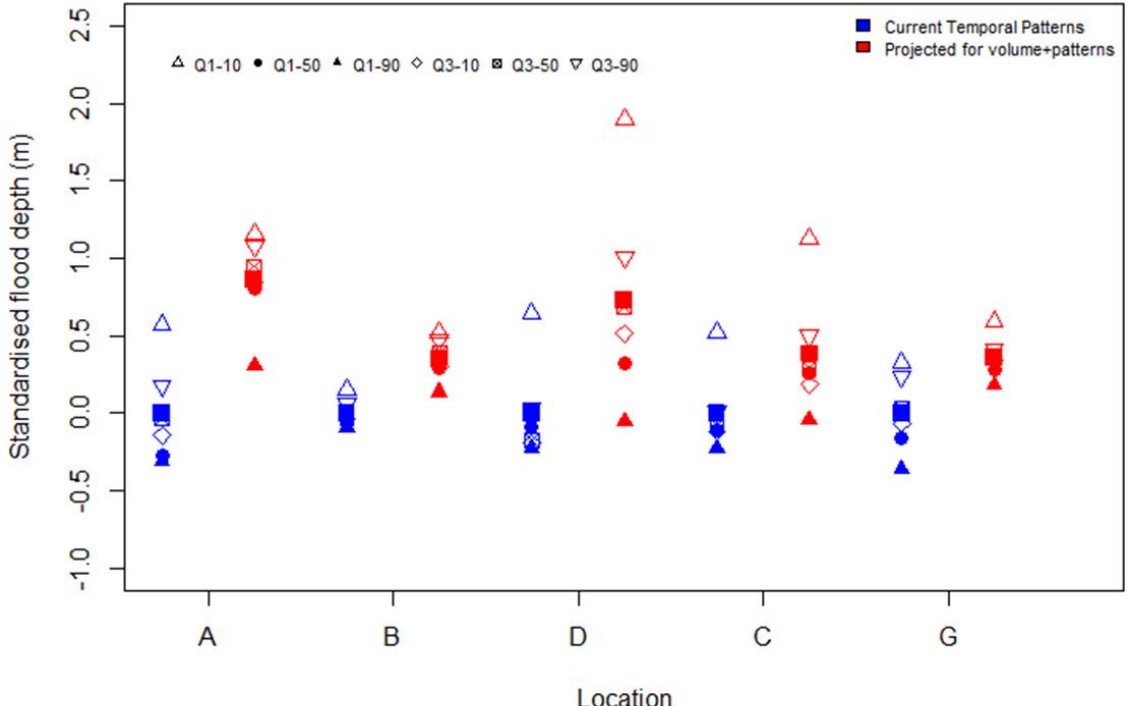


**541    Figure 9. Impact of rise in temperature on the peak flood depth variation at reference locations within the**

**542    watershed, when scaling is applied to both rainfall volume and temporal pattern.  The peak flood depths**

**543    at each reference point are based on 210 mm of total rain distributed over the 6 temporal patterns used.**

**544    Flood depths were standardised by subtracting the mean from the base simulations presented in Figure 6**

**545    for each location.**

As expected, substantial increase in flood risk is seen when the cumulative impacts of
changes to temporal pattern and increase in precipitation volume due to temperature rise are
modelled.   The mean flood depth is outside the upper margin of the highest flood depth for
base conditions except at the business park (C).  The business park location (C) comes close
to meeting this threshold as well.   The mean flood depth at Wilmes Lake (A) increases by
approximately 1 m, which translates to a significant increase in the extent of flooding. The
biggest change due to cumulative impacts occurs at the upstream location (B) were
previously, when only the temporal patterns were scaled, minimal impact was shown.   The
increase in flood depth at the reference locations due to changes to temporal patterns alone
range from 1 % to 35 %, while the cumulative impacts increase flood depth from 10 % to as
much as 170 %.   These results are similar to Zhou et al. (2016) who projects a 52% increase
in urban flooding for an RCP 8.5 scenario in China.   When considering all the nodes in the
model, the average increase in flood depth due only to changes in temporal patterns was 6 %.
The average increase in flood depth throughout the entire model due to cumulative impacts of
both changes to temporal pattern and rainfall volume is 37 %.   The percentage increase
(Table S2) shows that there is a significant impact to overall flood risk throughout the
catchment and that it is not isolated to the reference points that are discussed in detail.   These
results clearly show the increasing trend along with the significant variability in flood risk in
developed environments.
Additionally, the range of the results and hence the overall variability has increased at the
commercial and business park areas (C, D) locations when compared to Figure 8.   But this
change in the range is not consistent throughout the catchment. The higher intensity and the
larger total volume of rainfall overwhelm the existing infrastructure with much larger surface
overflows in different ways depending on the site and extents.   Also, the amount of increase
in the flood depths can change at different locations as the flooding increases.   The changes
to the range of depths as seen in Figure 9 suggests that quantifying and  accounting for
uncertainty in flood forecasts becomes more complex for future climates.
The use of detailed hydrologic and hydraulic modelling provides some of the nuances in
catchment response that adds important details to the results and our understanding on the
impacts of temporal patterns to flood risk, such as higher intensity rainfall does not always
results in the higher flood risk.   The variation of reference locations selected for this study
provides a reasonable assessment of how the flows interact with the physical features of the
catchment and how the results differ based on the location and features.   This study clearly
shows the sensitivity of the catchment to variation in how it rains, in particular the areas that
are more impacted by volume as opposed to flow rate. Explicitly including intensification of
rainfall patterns and volume due to climate change along with detailed H/H modelling to
assess the variability in catchment response makes this study unique among available
literature. The methodology presented here is universally applicable and the benefits of
correctly designing infrastructure are likely to far outweigh the cost of the added effort, even
in industry applications.

## 6     Conclusions

The significance of temporal patterns and how climate change impacts on rainfall patterns
affect flooding in developed environments was investigated using detailed hydrologic and
hydraulic modelling. Climate change impacts were undertaken by projecting historical
precipitation-temperature sensitivities on storm volumes and temporal patterns. The
following conclusions can be drawn from the results presented;

1. The response of a complex catchment is sensitive to variability in rainfall temporal pattern. The flood depths varied in excess of 1 m at Wilmes Lake when different temporal patterns were used with a constant volume of precipitation.

2. The variability of peak flood depth due to temporal pattern had similar magnitude when compared to variability due to total rainfall volume, which clearly shows that the temporal pattern of rainfall, or "how it rains" is as important as the volume of rainfall or "how much it rains" for the purposes of H/H modelling.

3. Temporal patterns add a quantifiable variability to the results generated in H/H modelling and need to be carefully consider when presenting results and associated uncertainties.

4. The temporal patterns intensified when scaled based on estimated temperature increases due to climate change.

5. A 1 % to 35 % increase in flood depth resulted when the scaled temporal patterns were used in the H/H model, suggesting an increase in potential flood risk purely due changes to "how it rains" as a result of climate change impacts.

6. A 10 % to 170 % increase in flood depth resulted when the projected rainfall volume was added to the projected temporal patterns, which shows a substantial increase in flood risk as a results climate change impacts on rainfall.

7. The variability of flood depth increased after temporal patterns and rainfall volumes were projected suggesting that H/H modelling for future planning and design needs to give serious consideration to the aspects of variability of rainfall patterns as well as increase in rainfall amounts.

8. Regional storage facilities are sensitive to rainfall patterns that are loaded at the latter

part of the storm duration while the extremely intense storms will cause flooding at all

locations.

The effect of projected intensification of storms due to climate change impacts suggests that
action needs to be taken promptly to prevent flood damages and possible loss of life. The two
most important points that can be derived from this study is that temporal patterns and storm
volumes need to be adjusted to account for climate change when applying to models of future
scenarios. The general application of H/H modelling analysis needs to adopt an ensemble
approach rather than a single event model to consider the significant variability in rainfall
patterns that can generate a substantial range in results in order to make a properly informed
decision as demonstrated here.
**Acknowledgments**
The authors acknowledge and thank the South Washington Watershed District
(http://www.swwdmn.org) in Minnesota, USA for providing the model as well as the
background data used for the analysis. We also acknowledge financial support of the
Australian Research Council.  The rainfall and temperature data for Minneapolis Airport and
locations around the site were taken from https://www.ncdc.noaa.gov/cdo-web/. NOAA Atlas
14 Volume 8 is available at http://www.nws.noaa.gov/oh/hdsc/PF_documents
Technical details of EPA-SWMM can be found at https://www.epa.gov/water-
research/storm-water-management-model-swmm.

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
