# Peer review of "Increase in flood risk resulting from climate change in a developed urban watershed–"

_Hydrology and Earth System Sciences, 2017_

## Referee Comment (RC1) · Anonymous Referee #1 · 25 Jul 2017

This manuscript addresses the impact of the changes in temporal pattern and volume of rainfall due to climate change on urban floods. In addition to the impact of total change in rainfall, the impact of the projected changes in temporal patterns alone is estimated. The background scientific question is important and the results are interesting. However, there are several issues that should be addressed before it is published.

<Major comments> - My major concern is the applicability of the scaling methods (both for volume and temporal pattern) for estimating the "projected" changes in the rainfall. The scaling factors are based on the relationship between the rainfall and temperature in the present climate. However, the present manuscript uses the scaling factor to

estimate the "projected" changes in the rainfall induced by the climate change. Both the temporal pattern and rainfall volume will be affected by the changes in various dynamic and thermodynamic facotors, not only by the changes in temperature. The applicability of the scaling method, which is based on the present climate variability, to the estimation of changes in rainfall under climate change should be verified. At least, it should be discussed in the manuscript.

L231-232: The characteristics of the temporal pattern selected from NOAA ATLAS is important in this study. It should be explained more in the manuscript or figures about what the six temporal patterns are like.

<Minor comments> L224-226: How the spatial distribution of rainfall in the catchment considered? Is it uniform over the cathment? Please describe it in the manuscript.

Table 2 (Design Rainfall): Why don't you use the same unit (e.g., mm/24hour) for all three rainfalls?

Table 2 (descriptions of temporal patterns): I don't understand what the "1st quantile 10th percentile" is. Explaining more about the temporal pattern will help reader's better understanding. To show the shape of the pattern in the figure my be helpful.

L264-270: Using some equations for the explanation on the volume scaling may be helpful for readers.

L399 ".. as shown in Figure 5(a)": Should be Figure 6(a)?

L443-445 "...the mean of the flood depth for projected events does exceed the upper limit of the variability in flood depths for the base scenario": I don't know which part of the figure shows the upper limit of the variability for the base scenario.

L477-478 "The increase .. due to changes to temporal patterns alone range from 1% to 35%": Does these persentage numbers come from Figure 7? Since the unit of the Figure 7 is meter, it is difficult to figure out the percentage change from Figure 7.

---

## Referee Comment (RC2) · Anonymous Referee #2 · 28 Jul 2017

This manuscript entitled "Increase in urban flood risk resulting from climate change – The role of storm temporal patterns" draws readers' attention towards importance of storm temporal pattern in urban flood modeling under altering climatic scenario. Given the frequent reporting of urban floods across the globe this study provides useful insight to urban flood modelers. The manuscript fits the aim and scope of HESS quite well, and can be accepted provided authors address the following comments with required modifications and justifiable responses.
* * *
Major Comments:

[Figure]

Comment 1:

Why did authors choose a 50-year return period storm? Why not 10, 20, or 25 years return period that is much common for urban flood modeling studies? Or why not 100 year return period?

Comment 2:

RCP 8.5 scenario is derived using the most pessimistic assumption and is very unlikely given the ongoing worldwide efforts to curb the carbon emission and green initiatives. Though such studies using RCP 8.5 gives mind boggling figures, these remain very unlikely. A more likely scenario could be RCP 4.5 should have been used along RCP 8.5 to encompass the effects of climatic change. Secondly, authors carried out the study for projected period for 2081-2100 skipping the intermediate time frames. Is there no significant results during 2025-2050 or 2051-2080? Though the results would be much pronounce in the later part of the century, intermediate time frame should also be discussed. Authors must explain the rationale behind selecting worst case climatic scenario i.e., RCP 8.5 and also come up with the reasoning to skip RCP 4.5 and selection of specific time frames for such modeling exercises for potential users. Additional details related to exercise can be provided in supplementary materials.

Comment 3:

The study employs the modeling component in a big way to derive the conclusions, however, there is no discussion made on how the modeling framework was setup. Catchment sizes in the modeling setup varies from 0.25 sq km to 22 sq km that makes almost 90 times change in smallest and largest catchment. Interestingly, unlike river basin scale studies in urban drainage modeling catchment boundaries are not demarcated by their natural topography as the interceptor drains divert the runoff water omitting the natural stream lines. How the authors have discretized such vastly different sized catchments?

Authors should discuss how the impervious area is estimated to include in modeling framework, and other parameters used in the modeling exercise should be tabulated. Did authors fed the existing storm sewage network into the model to rout the flow from a particular sub-catchment to outlet or directed them directly to the outlet from the sub-catchment? Also discuss how the model was calibrated and validated. A separate section on model setup is highly warranted to make the manuscript more informative.

Comment 4:

Line 266-267 and Figure 4: "The rainfall-temperature pairs were binned on 2 degree temperature bins . . ."

Does it mean that binning was done by counting the number of rainfall events and their corresponding magnitudes at each 2 degree temperature interval? What does the height of each bin depict? What do the count and precipitation magnitudes from primary and secondary y-axis show?

Comment 5:

In Line 339-340 authors say "The flood depths extracted from the model were first analyzed to compare variability between temporal patterns and total rainfall depth. . ."

SWMM is a 1-dimensional model and does not simulate the flood extent or flood depth. Though it simulates depth of water being flooded from a node, it depends on the adequacy of drainage network. While discussing the flood depth in relation to urban scenario, the depth of flood inundation should be used rather that the depth of total water flooded from a particular node or from the entire system. This aspect need some clarification.

Comment 6:

In Line 342-343 authors say "These sub-models show the variation in catchment response to runoff generated by different land use types. . ."

[Figure]

There is no provision of feeding LULC information in SWMM, rather it takes percent pervious and impervious area. Different land use types gives a notion that model is simulating overland flow explicitly for residential, paved surfaces, parks, grassed land etc. How the different land use land cover type were incorporated in the model?

Similarly, in Line 360 and 368 authors talk about "local storage/ local natural storage". How these storage was incorporated into the modeling exercise.

Comment 7:

Line 399-411 does not helps much and as a reader I find it less convincing how Fig 6(a) is different than Fig. 6(b) and how pronounce the difference is for temporal pattern case and total rainfall volume case. Moreover, visually Figure 6(a) and (b) are seems more or less identical with little change. It would be better if author can redraw them to convey their point. Perhaps, comparison of Q1-50 and Q3-50 in same graph for temporal pattern variation or total rain volume variation will help the readers' understanding. Also specific markers for different cases should be provided, as of now there are 4 squares and each belong to which requires a thorough reading. Make the image self-explanatory.

Comment 8:

Fourth conclusion suggests the 'increase in potential flood risk purely due changes to "how it rains" as a result of climate change impacts'.

This conclusion is drawn from the analysis shown in Fig 6 and Fig. 7. How the temporal pattern variation has a pronounce effect on flood risk as from the Fig. 6 gives almost same picture for temporal pattern for Q1 and Q3 rainfall, whereas from Fig. 7 also not much significant change can be noticed in the standardized flood depth due to current temporal patterns and projected patterns unlike Fig. 8, where the difference is really remarkable. An elaboration would help the readers' understanding.

Comment 9:

Temporal pattern or distribution used from NOAA ATLAS should be discussed in short. It's not clear what does nth quartile at mth percentile means. It would be insightful if authors show it in figure.

Comment 10:

In Line 283, what does author mean by "current industry standard temporal distributions"?:

Authors may like to use supplementary material space for elaborate discussion to clarify the doubt.
* * *
Minor Comments:

(1) First line of abstract [Line 8-9] i.e., "Warming temp . . ." is almost repeated in [Line 18-19] i.e., "Current literature . . ."

(2) Fix the citation formats throughout the text, for example in [Line 89] the citation should be like Milly et al. (2007).

(3) Delete 'an' before EPA-SWMM in [Line 182], delete '2016' after EPA in [Line 185]

(4) Line 64: Correct the "Intensity/Duration/Frequency" as "Intensity-Duration-Frequency"

(5) Line 114-116: It would not be apt to link Uttarakhand and Kashmir floods in India with poor stormsewer design from Bisht et al. (2016). As these floods were caused by cloud burst and moreover the topography is hilly in that place. However, Bisht et al. (2016) discussed the Mumbai flood that can be aptly link with flood risk caused by inadequate storm drainage.

(6) Line 165-168: These line should come in the last of introduction section where authors generally list down the objectives or novelty of their work.

(7) Line 231-232: Cite the NOAA ATLAS like any other technical report and list in the reference. Table 2: Use consistent unit for all the design rainfall.

(8) Line 291: There is no reference for Figure SPM7(a)(IPCC 2014) in reference section. This Figure can be adopted from the source in the manuscript.

(9) Figure 1: What do those lines in Orange, magenta, and Black depict? Proper legends discussing each feature must be included with the figure to make it meaningful. The backdrop can be removed as it is making the image complex to understand.

(10) Figure 6: Figure caption can be shortened as "Comparison of total volume of rainfall and temporal patterns variability impact on peak flood depth. Flood depth variation due to the 6 different temporal patterns with 160 mm of rain compared to 110, 160 and 210 mm of total rainfall over 24 hours distributed over (a) Q1-50 temporal pattern (b) Q3-50 temporal pattern. Flood depths were standardised by subtracting the mean at each location for ease of comparison"

(11) Figure 7: Increase the font size of legends.

―――――――――――――――――

---

## Author Comment (AC1) · 11 Sep 2017

**Response to Reviewer 1**

This manuscript addresses the impact of the changes in temporal pattern and volume of rainfall due to climate change on urban floods. In addition to the impact of total change in rainfall, the impact of the projected changes in temporal patterns alone is estimated. The background scientific question is important and the results are interesting. However, there are several issues that should be addressed before it is published.

We thank the reviewer for their time and positive assessment of the manuscript. We address the reviewer's concerns in turn with our responses in italics. Please note that the author comment (AC) and Proposed Changes in Manuscript (PCiM) based on the comments are indicated as such separately for each comment.

**Major comments**

**MC1R1**

My major concern is the applicability of the scaling methods (both for volume and temporal pattern) for estimating the "projected" changes in the rainfall. The scaling factors are based on the relationship between the rainfall and temperature in the present climate. However, the present manuscript uses the scaling factor to estimate the "projected" changes in the rainfall induced by the climate change. Both the temporal pattern and rainfall volume will be affected by the changes in various dynamic and thermodynamic factors, not only by the changes in temperature. The applicability of the scaling method, which is based on the present climate variability, to the estimation of changes in rainfall under climate change should be verified. At least, it should be discussed in the manuscript.

AC- In this work we assume that temperature is the primary climatic variable associated with changing rainfall extremes and have adopted a scaling of 4.7% per degree Celsius. This value appears to be consistent both with historical trends and climate change projections.

Figure 4c of Barbero et al (2017) looks at a non-stationary extreme value analysis and finds a sensitivity of approximately 7%/oC for a non-stationary Theil-Sen estimator for North America. Globally, Westra et al., (2013) find historical trends have global sensitivity between 5.9%/oC and 7.7%/oC. However, Kharin et al (2013) report an approximately 4% sensitivity over land globally from the CMIP5 model results with a range of 2.5-5% for the U.S.A.

In regard to the evidence above we believe our projections are consistent with the available evidence regarding precipitation change. There is a possibility that it slightly underestimates historical trends, but is at the upper end of predictions from climate model predictions.

Finally, there are a number of published works which show that temperature is a recommended covariate for projecting rainfall e.g. Agilan and Umamahesh (2017) and Ali and Mishra (2017) may indeed implicitly account for dynamic factors (Wasko and Sharma (2017).

PCiM- The above discussion on sensitivity will be added at line 316 to help the reader evaluate the scaling used. We note that the value of 2.92 at Line 310 is a typo and indeed should be 4.7% (as shown correctly in Figure 4).

A discussion regarding the validity of using temperature as a covariate for projecting rainfall would be included in the modified manuscript at line 263 to expand on the justification of using temperature scaling beyond the reference to Lenderink and Attema (2015).

**MC2R1**

L231-232: The characteristics of the temporal pattern selected from NOAA ATLAS is important in this study. It should be explained more in the manuscript or figures about what the six temporal patterns are like.

AC- We agree with this comment. The quartiles indicate the timing of the greatest percentage of total rainfall that occurs during a storm. First quartile would indicate that the majority of the rainfall including the peak will occur in the  $1^{st}$  ¼ of the duration, which is between hours 1 through 6 in the case of a 24-hour storm (As indicated in chart a). The distributions were further analysed to determine the frequency of occurrence within each quartile to determine a percentile for each distribution.

PCiM-Figure R1 will be added to the manuscript which shows the different patterns that were used in this manuscript. Further, the following text will be added at line 243-'The quartiles indicate the timing of the greatest percentage of total rainfall that occurs during a storm. First quartile would indicate that the majority of the rainfall including the peak will occur in the 1st ¼ of the duration, which is between hours 1 through 6 in the case of a 24-hour storm. The temporal distributions were also separated in Atlas 14 to determine the frequency of occurrence within each quartile to determine a percentile for each distribution." Will also add reference to Figure 4 in Appendix A5 of NOAA Atlas 14.

Figure R1. NOAA Atlas 14 temporal patterns used in the modelling

**Reviewer 1 Minor Comments**

C1R1

L224-226: How was the spatial distribution of rainfall in the catchment considered? Is it uniform over the catchment? Please describe it in the manuscript.

**AC-Yes, the rainfall is assumed to be uniform over the catchment.**

*PCiM-* A statement on the spatial distribution of rainfall will be added in the manuscript at Line 240.

**C2R1**

Table 2 (Design Rainfall): Why don't you use the same unit (e.g., mm/24hour) for all three rainfalls?

AC- We agree and thank you for catching that oversight. We will correct the table to ensure all rainfalls appear in mm.

PCiM- the table will be updated to;

**Table 2. Description of notation used in reference to the modelled storm depths andtemporal distributions (NOAA Atlas 14 volume 8 appendix 5)**

| Design Rainfall | Description                                                           |
|-----------------|-----------------------------------------------------------------------|
| 160 mm 24 hour  | 2 % exceedance 24-hour duration (50-year return period) rainfall      |
|                 | depth                                                                 |
| 125 mm 24 hour  | Lower margin of the 90% confidence interval of the 2 % exceedance     |
|                 | 24-hour duration (50-year return period) rainfall depth-Approximately |
|                 | Equivalent to the 20-year 24 hour ARI                                 |
| 210 mm 24 hour  | Upper margin of the 90% confidence interval of the 2 % exceedance     |
|                 | 24-hour duration (50-year return period) rainfall depth-Approximately |
|                 | Equivalent to the 200-year 24 hour ARI                                |

**C3R1**

Table 2 (descriptions of temporal patterns): I don't understand what the "1st quantile 10th percentile" is. Explaining more about the temporal pattern will help reader's better understanding. To show the shape of the pattern in the figure may be helpful.

AC-As explained in the response above, the quartiles indicate the timing of the greatest percentage of total rainfall that occurs during a storm. First quartile would indicate that the majority of the rainfall including the peak will occur in the 1st ¼ of the duration, which is between hours 1 through 6 in the case of a 24-hour storm. The percentile indicates the frequency of occurrence of each pattern within each quartile. In general, the percentile indicates the level of intensity within each quartile with a lower percentile referring to a higher intensity and a lower probability of occurrence. The reviewer is right that this was not adequately explained in the original manuscript and this discussion will be added to the paper.

PCiM- Please refer to response in comment MC2R1

**C4R1**

L264-270: Using some equations for the explanation on the volume scaling may be helpful for readers.

AC-Reviewer 2 also commented on the relatively short explanation of the methodology. The text at lines 264-270 will be expanded as per below:

**PCiM- Lines 264-270 will be expanded to include;**

"Using established methods (Hardwick Jones et al., 2010a; Utsumi et al., 2011; Wasko and Sharma, 2014), the volume scaling for the 24 hour storm duration was calculated using an exponential regression. The results are presented in Figure 4. First, daily rainfall was paired with daily average temperature. The rainfall-temperature pairs were binned on 2°C temperature bins, overlapping with steps of one degree. For each 2°C bin a Generalized Pareto Distribution fitted to the rainfall data in the bin that was above the 99th percentile to find extreme rainfall percentiles (Lenderink et al., 2011; Lenderink and van Meijgaard, 2008). Extreme percentiles below the 99th percentile (inclusive) were calculated empirically. A linear regression was subsequently fitted to the fitted log-transformed extreme percentiles and used as the rainfall volume scaling (Figure 4). Hence the volume (V) is related to a change in temperature (T) by

$$V_2 = V_1 (1 + \alpha)^{\Delta T}$$

Where  $\alpha$  is the scaling of the precipitation per degree change in temperature."

C5R1

L399 ".. as shown in Figure 5(a)": Should be Figure 6(a)?

AC- The Figure reference in the manuscript is correct as is. The intent is to show that the results at Location A are similar.

PCiM- Will change sentence to replace 'as shown' with "similar to results shown in".

C6R1

L443-445 "...the mean of the flood depth for projected events does exceed the upper limit of the variability in flood depths for the base scenario": I don't know which part of the figure shows the upper limit of the variability for the base scenario.

**AC-We agree that this statement was a bit vague.**

PCiM- This sentence will be re-written to read: "the mean of the flood depth for projected events (shown in red Figure 7) exceeds the upper limit of the variability, or spread, of flood depths for the base scenario (shown in blue in Figure 7)."

C7R1

L477-478 "The increase .. due to changes to temporal patterns alone range from 1% to 35%": Does these percentage numbers come from Figure 7? Since the unit of the Figure 7 is meter, it is difficult to figure out the percentage change from Figure 7.

AC-The percentages are based on the results that were used to generate Figures 6 and 7. Tables showing percentage calculations will be added to the supplemental information to make these results more clear.

| Impact on flood depth from projected temporal pattern |    |    |     |     |     |     |     |  |  |  |
|-------------------------------------------------------|----|----|-----|-----|-----|-----|-----|--|--|--|
|                                                       | А  | В  | С   | Е   | D   | F   | G   |  |  |  |
| Q1-10                                                 | 1% | 1% | 1%  | 0%  | 2%  | 3%  | 0%  |  |  |  |
| Q1-50                                                 | 3% | 3% | 20% | 8%  | 8%  | 1%  | 7%  |  |  |  |
| Q1-90                                                 | 0% | 0% | 12% | 10% | 13% | 1%  | -1% |  |  |  |
| Q3-10                                                 | 4% | 5% | 19% | 10% | 16% | 2%  | 9%  |  |  |  |
| Q3-50                                                 | 5% | 8% | 18% | 10% | 35% | 2%  | 9%  |  |  |  |
| Q3-90                                                 | 4% | 5% | 16% | 7%  | 12% | 15% | 1%  |  |  |  |

**PCiM- The following tables will be added in Supplementary Information**

Impact on flood depth from projected volume and temporal pattern

|       | А   | В   | С   | D    | E   | F    | G   |
|-------|-----|-----|-----|------|-----|------|-----|
| Q1-10 | 13% | 40% | 53% | 95%  | 21% | 108% | 12% |
| Q1-50 | 31% | 51% | 74% | 69%  | 37% | 57%  | 25% |
| Q1-90 | 17% | 34% | 49% | 37%  | 27% | 9%   | 35% |
| Q3-10 | 27% | 46% | 61% | 147% | 59% | 76%  | 22% |
| Q3-50 | 26% | 52% | 68% | 173% | 57% | 119% | 17% |
| Q3-90 | 23% | 48% | 78% | 140% | 49% | 170% | 8%  |

**References:**

Ali, H., and V. Mishra (2017), Contrasting response of rainfall extremes to increase in surface air and dewpoint temperatures at urban locations in India, Sci. Rep., 7(1), 1228, doi:10.1038/s41598-017-01306-1.

Agilan, V., and N. V Umamahesh (2017), What are the best covariates for developing nonstationary rainfall Intensity-Duration-Frequency relationship?, Adv. Water Resour., 101, 11– 22, doi:10.1016/j.advwatres.2016.12.016.

Barbero, R., H. J. Fowler, G. Lenderink, and S. Blenkinsop (2017), Is the intensification of precipitation extremes with global warming better detected at hourly than daily resolutions?, Geophys. Res. Lett., 1–10, doi:10.1002/2016GL071917.

Kharin, V. V., F. W. Zwiers, X. Zhang, and M. Wehner (2013), Changes in temperature and precipitation extremes in the CMIP5 ensemble, Clim. Change, 119(2), 345–357, doi:10.1007/s10584-013-0705-8.

Westra, S., L. Alexander, and F. Zwiers (2013), Global increasing trends in annual maximum daily precipitation, J. Clim., 26, 3904–3918, doi:http://dx.doi.org/10.1175/JCLI-D-12-00502.1.

---

## Author Comment (AC2) · 11 Sep 2017

This manuscript entitled "Increase in urban flood risk resulting from climate change – The role of storm temporal patterns" draws readers' attention towards importance of storm temporal pattern in urban flood modelling under altering climatic scenario. Given the frequent reporting of urban floods across the globe this study provides useful insight to urban flood modellers. The manuscript fits the aim and scope of HESS quite well, and can be accepted provided authors address the following comments with required modifications and justifiable responses.

*We thank the reviewer for their time. We also thank the reviewer for their favourable assessment of the manuscript content and results presented. We address the reviewers concerns in turn with our responses in italics. Please note that the author comment (AC) and Proposed Changes in Manuscript (PCiM) based on the comments are indicated as such separately for each comment.*

**Major comments**

MC1R2

Why did authors choose a 50-year return period storm? Why not 10, 20, or 25 years return period that is much common for urban flood modelling studies? Or why not 100 year return period?

*AC-The model was setup in year 2000 and has been continuously updated and maintained over the 15+ years based on the prevailing 100-year 24-hour rainfall event for that catchment. Atlas 14 V8 has updated the rainfall depths such that the previous 100-yr rainfall depth is now the 50-year rainfall depth. Since this paper looks at relative impacts to peak flood depths based on rainfall patterns and future climate conditions and does not project absolute flood depths, the authors selected to use the old 100-yr rainfall as the base rainfall event with the correct current reference of a 50 year event. The substantial effort to update the model to accommodate the new 100-year rainfall depth would not have provided an incremental value in the results.*

MC2R2

RCP 8.5 scenario is derived using the most pessimistic assumption and is very unlikely given the ongoing worldwide efforts to curb the carbon emission and green initiatives. Though such studies using RCP 8.5 gives mind boggling figures, these remain very unlikely. A more likely scenario could be RCP 4.5 should have been used along RCP 8.5 to encompass the effects of climatic change. Secondly, authors carried out the study for projected period for 2081-2100 skipping the intermediate time frames. Is there no significant results during 2025-2050 or 2051-2080? Though the results would be much pronounce in the later part of the century, intermediate time frame should also be discussed. Authors must explain the rationale behind selecting worst case climatic scenario i.e., RCP 8.5 and also come up with

the reasoning to skip RCP 4.5 and selection of specific time frames for such modelling exercises for potential users. Additional details related to exercise can be provided in supplementary materials.

*AC-As suggested, the authors did an additional analysis for the case of RCP 4.5, looking at a temperature change of $3^0C$ . The following table illustrates that the trend in results are similar to the RCP 8.5 scenario for all cases as expected. Understandably, the impact to flood depths is not as significant as when looking at a $5^0C$ increase in temperature. The main goal of the paper is to demonstrate the importance of accounting for the changes expected in temporal patterns of rainfall, which looks at relative impacts.*

*We wish to note that there is literature suggesting that we are tracking on a RCP8.5 scenario (Peter. G et al 2013) and indeed many forecasts suggest greater temperature increases over land. The authors agree that there should be rigorous thought on how far out and what level of climate impacts should be considered when selecting a threshold for design or when setting absolute flood depths.*

*PCiM- The following table will be added in Supplementary Information with explanatory narrative.*

*Table S1- Results showing normalized flood depths around the mean at each location for a projected temperature increase of 3 deg C.*

|  | Current (normalized) | | | | |
| --- | --- | --- | --- | --- | --- |
|  | **A** | **B** | **D** | **C** | **G** |
| Q1-10 | 0.58 | 0.15 | 0.65 | 0.53 | 0.33 |
| Q1-50 | -0.28 | -0.07 | -0.08 | -0.11 | -0.16 |
| Q1-90 | -0.31 | -0.10 | -0.23 | -0.24 | -0.37 |
| Q3-10 | -0.14 | -0.04 | -0.19 | -0.12 | -0.07 |
| Q3-50 | -0.03 | -0.01 | -0.18 | -0.08 | 0.03 |
| Q3-90 | 0.18 | 0.07 | 0.03 | 0.01 | 0.24 |

|  | Projected patterns (normalized) | | | | |
| --- | --- | --- | --- | --- | --- |
|  | **A** | **B** | **D** | **C** | **G** |
| Q1-10 | 0.60 | 0.16 | 0.67 | 0.54 | 0.33 |
| Q1-50 | -0.21 | -0.06 | -0.05 | -0.04 | -0.09 |
| Q1-90 | -0.31 | -0.10 | -0.19 | -0.21 | -0.38 |
| Q3-10 | -0.05 | -0.02 | -0.15 | -0.06 | 0.03 |
| Q3-50 | 0.11 | 0.02 | -0.07 | -0.02 | 0.16 |
| Q3-90 | 0.27 | 0.10 | 0.08 | 0.06 | 0.25 |
| *Change in Mean* | *0.07* | *0.02* | *0.05* | *0.04* | *0.05* |

| | Projected patterns and volumes (normalized) | | | | |
|---|---|---|---|---|---|
| | **A** | **B** | **D** | **C** | **G** |
| Q1-10 | 0.98 | 0.38 | 1.36 | 0.73 | 0.49 |
| Q1-50 | 0.34 | 0.14 | 0.04 | 0.09 | 0.25 |
| Q1-90 | 0.04 | 0.03 | -0.15 | -0.15 | -0.09 |
| Q3-10 | 0.47 | 0.16 | 0.15 | 0.03 | 0.27 |
| Q3-50 | 0.91 | 0.36 | 0.60 | 0.23 | 0.34 |
| Q3-90 | 0.91 | 0.36 | 0.60 | 0.23 | 0.34 |
| *Change in Mean* | *0.61* | *0.24* | *0.44* | *0.19* | *0.27* |

MC3R2

The study employs the modelling component in a big way to derive the conclusions, however, there is no discussion made on how the modelling framework was setup. Catchment sizes in the modelling setup varies from 0.25 sq km to 22 sq km that makes almost 90 times change in smallest and largest catchment. Interestingly, unlike river basin scale studies in urban drainage modelling catchment boundaries are not demarcated by their natural topography as the interceptor drains divert the runoff water omitting the natural stream lines. How the authors have discretised such vastly different sized catchments?  Authors should discuss how the impervious area is estimated to include in modelling framework, and other parameters used in the modelling exercise should be tabulated. Did authors fed the existing storm sewage network into the model to rout the flow from a particular sub-catchment to outlet or directed them directly to the outlet from the subcatchment? Also discuss how the model was calibrated and validated. A separate section on model setup is highly warranted to make the manuscript more informative.

*AC- As mentioned in the response above, the model used in the study was set initially in year 2000 and has been continuously maintained and updated to include the latest available landuse/landcover and stormwater infrastructure information.  The model includes all components of stormwater conveyance within the catchment including sewers, open channels and storage areas, along with street overflows.  Highly detailed delineation of both sub-catchment boundaries and impervious area was done using a high resolution DEM, development construction and grading plan overlays and aerial imagery within a GIS environment.  All surface runoff is fed into the appropriate inflow points of the hydraulic conveyance system.  The model has been validated and used to design major capital improvement and flood mitigation projects over the years.  The following link connects to a report that discusses extensive model validation work based on an extreme storm. http://www.swwdmn.org/pdf/projects/completed/2006%20Stormwater%20Modeling%20Report_HDR.pdf*

*PCiM – A paragraph on the model build as explain above will be added at line ###.  The following references will be added in the manuscript as citations.*

*Model Development and related information:*

*Hettiaracchchi, S, and W. Johnson (2006), Stormwater modelling Report, HDR Project No. 32072, [available online:*
*http://www.swwdmn.org/pdf/projects/completed/2006%20Stormwater%20Modeling%20Report_HDR.pdf]*

*The following is an additional publication that discusses flood mitigation projects analysed using this model.*

*Hettiarachchi. S, Beduhn. R, Christopherson. J Moore. M, Managing Surface Water for Flood Damage Reductio, World Water and Environmental Resources Congress 2005, May 15-19, 2005 | Anchorage, Alaska, United States, doi-10.1061/40792(173)321*

*Many of the projects listed here are based on using this model.*

*http://www.swwdmn.org/projects/*

MC4R2

Line 266-267 and Figure 4: "The rainfall-temperature pairs were binned on 2 degree temperature bins . . ." Does it mean that binning was done by counting the number of rainfall events and their corresponding magnitudes at each 2 degree temperature interval? What does the height of each bin depict? What do the count and precipitation magnitudes from primary and secondary y-axis show?

*AC-The first reviewer also commented on the clarity of this paragraph. We believe some confusion arose from the histogram in Figure 4 and having two sets of axes. The histogram would have effectively only shown every second bin (as the binning is performed using two degree bins with overlapping steps of one degree).*

*PCiM- Lines 264-270 will be expanded to read as follows and Figure 4 will be replaced with the figure below:"Using established methods (Hardwick Jones et al., 2010a; Utsumi et al., 2011; Wasko and*
*Sharma, 2014), the volume scaling for the 24 hour storm duration was calculated using an exponential regression. The results are presented in Figure 4. First, daily rainfall was paired with daily average temperature. The rainfall-temperature pairs were binned on $2°C$ temperature bins, overlapping with steps of one degree. For each $2°C$ bin a Generalized Pareto Distribution fitted to the rainfall data in the bin that was above the 99th percentile to find extreme rainfall percentiles (Lenderink et al., 2011; Lenderink and van Meijgaard, 2008). Extreme percentiles below the $99^{th}$ percentile (inclusive) were calculated empirically. A linear regression was subsequently fitted to the fitted log-transformed extreme percentiles and used as the rainfall volume scaling (Figure 4). Hence the volume (V) is related to a change in temperature (T) by*

$$V_2 = V_1(1 + \alpha)^{\Delta T}$$

*Where α is the scaling of the precipitation per degree change in temperature."*

*In light of the above comments Figure 4 has been modified and the new figure is shown below. The histogram in the figure has been removed to prevent confusion as the fitted quantiles were not necessarily matching the histogram bins presented creating ambiguity in the results.*

[Figure]

*Figure 4 Scaling total volume of rainfall with temperature for Minneapolis (1901-2014 daily rainfall). Grey dots are rainfall temperature-pairs and the coloured dots are the extreme percentiles. The grey dashed line represents a scaling of 7 %.*

MC5R2

In Line 339-340 authors say "The flood depths extracted from the model were first analysed to compare variability between temporal patterns and total rainfall depth. . ." SWMM is a 1-dimensional model and does not simulate the flood extent or flood depth. Though it

simulates depth of water being flooded from a node, it depends on the adequacy of drainage network. While discussing the flood depth in relation to urban scenario, the depth of flood inundation should be used rather that the depth of total water flooded from a particular node or from the entire system. This aspect need some clarification.

*AC- The review is correct in how SWWM typically shows flooded nodes and yes, the flood depth are dependent on the adequacy of the model. As discussed in the comment regarding the background and extent of the model build, extensive surface overflow routes from flooded areas as well as explicitly modelling street overflows and storage extents are included in the model geometry. This allows the resulting flood depths to take into account the flood extents as well, opposed to the typical funnel that SWMM uses at nodes. Additionally, majority of the reference locations are at local storage nodes that provide a good representation of flood extents. Storage nodes have depth/area curves that represent flood extents at each depth. Therefore, the results from this model provide a reasonably accurate representation of extents related to each flood depth.*

*PCiM- This discussion will be added to model discussion at line ##.*

MC6R2

In Line 342-343 authors say "These sub-models show the variation in catchment response to runoff generated by different land use types. . ." There is no provision of feeding LULC information in SWMM, rather it takes percent pervious and impervious area. Different land use types gives a notion that model is simulating overland flow explicitly for residential, paved surfaces, parks, grassed land etc. How the different land use land cover type were incorporated in the model? Similarly, in Line 360 and 368 authors talk about "local storage/ local natural storage". How these storage was incorporated into the modelling exercise.

*AC- Agree with the comment that SWMM does not have provisions to explicitly designate LULC in a runoff area. However, as discussed in response to comment 3, the model was setup in extensive detail using multiple layers of information that provide characteristic percentages of impervious area based on the built environment within each of the sub-catchments. By discretising to small areas, the model then is able to isolate the various landuse types within each catchment and generate a composite impervious percentage and a rate of runoff representative of each different landuse type. Local constructed storage refers to stormwater ponds that were built as part of rate and volume controls to meet post development rules requirements. Natural storage locations refer to existing ponds and lakes within the catchments. The storage information is added into the model as depth/area tables using the DEM and bathymetric survey (major storage locations) for natural storage locations and construction plans for constructed storage locations.*

*PCiM- N/A*

MC7R2

Line 399-411 does not helps much and as a reader I find it less convincing how Fig 6(a) is different than Fig. 6(b) and how pronounce the difference is for temporal pattern case and total rainfall volume case. Moreover, visually Figure 6(a) and (b) are seems more or less identical with little change. It would be better if author can redraw them to convey their point. Perhaps, comparison of Q1-50 and Q3-50 in same graph for temporal pattern variation or total rain volume variation will help the readers' understanding. Also specific markers for different cases should be provided, as of now there are 4 squares and each belong to which requires a thorough reading. Make the image self-explanatory.

*AC- We agree with the reviewer and we will redraw Figure 6 to convey the intended point. Figure 6 (a) and (b) attempts to illustrate the variation between temporal pattern vs volume of rainfall, and is not intended to show changes based on a particular, or each temporal pattern. The fact that Figure 6a and 6b are similar shows that this variability is generally independent of the temporal pattern chosen for the volume variability. That is to say, that the results are not skewed to be favourable by picking a single temporal pattern to examine the volume variability.*

*PCiM- Figure 6 (a) and (b) will be modified to emphasize the range of results as well as the comparison between current and projected results.*

*MC8R2*

Fourth conclusion suggests the 'increase in potential flood risk purely due changes to "how it rains" as a result of climate change impacts'. This conclusion is drawn from the analysis shown in Fig 6 and Fig. 7. How the temporal pattern variation has a pronounce effect on flood risk as from the Fig. 6 gives almost same picture for temporal pattern for Q1 and Q3 rainfall, whereas from Fig. 7 also not much significant change can be noticed in the standardized flood depth due to current temporal patterns and projected patterns unlike Fig. 8, where the difference is really remarkable. An elaboration would help the readers' understanding.

*AC-The reviewer is correct that the 4$^{th}$ conclusion is based on the data that was used to generate Figure 7. This conclusion points to flood impacts that are due to projected temporal patterns. Whereas Figure 8 shows flood impacts due to projection of both temporal patterns and rainfall volumes.*

*PCiM-We will elaborate further within the discussion to improve how we convey this point.*

MC9R2

Temporal pattern or distribution used from NOAA ATLAS should be discussed in short. It's not clear what does nth quartile at mth percentile means. It would be insightful if authors show it in figure.

*AC- We agree with this comment as well as the comment by the other reviewer. Figure R1 will be added to the manuscript which shows the different patterns that were used in this manuscript.*

*PCiM-Figure R1 will be added to the manuscript which shows the different patterns that were used in this manuscript. Further, the following text will be added at line 243-'The quartiles indicate the timing of the greatest percentage of total rainfall that occurs during a storm. First quartile would indicate that the majority of the rainfall including the peak will occur in the 1st ¼ of the duration, which is between hours 1 through 6 in the case of a 24-hour storm. The temporal distributions were also separated in Atlas 14 to determine the frequency of occurrence within each quartile to determine a percentile for each distribution." Will also add reference to Figure 4 in Appendix A5 of NOAA Atlas 14.*

.

[Figure]

*Figure R1. NOAA Atlas 14 temporal patterns used in the modelling*

MC10R2

In Line 283, what does author mean by "current industry standard temporal distributions"?: Authors may like to use supplementary material space for elaborate discussion to clarify the doubt.

*AC- Temporal patterns from design guidelines and standards are used throughout civil engineering and consulting industry for design flood estimation and these standards are commonly referred as 'industry standards'.*

*PCiM- To clarify what is meant by this the text at line 283 would be replaced with "temporal patterns for design flood estimation" instead of "current industry standard" and refer to the example of the NOAA Atlas 14 temporal patterns.*

**minor comments**

C1R2

First line of abstract [Line 8-9] i.e., "Warming temp . . ." is almost repeated in [Line 18-19] i.e., "Current literature . . ."

*AC- Agree and will remove the first sentence to prevent repetition.*

*PCiM- Edit first sentence as indicated.*

C2R2

Fix the citation formats throughout the text, for example in [Line 89] the citation should be like Milly et al. (2007).

*AC-Agreed and appreciate noting the need to adjust the citation.*

*PCiM- The citations will be edited appropriately so that the parenthesis occur in the correct location.*

C3R2

Delete 'an' before EPA-SWMM in [Line 182], delete '2016' after EPA in [Line 185]

*AC and PCiM- Thank you. Will perform these edits.*

C4R2

Line 64: Correct the "Intensity/Duration/Frequency" as "Intensity-Duration-Frequency"

*AC and PCiM-Will update paper to reflect suggested change*

C5R2

Line 114-116: It would not be apt to link Uttarakhand and Kashmir floods in India with poor storm sewer design from Bisht et al. (2016). As these floods were caused by cloud burst and moreover the topography is hilly in that place. However, Bisht et al. (2016) discussed the Mumbai flood that can be aptly link with flood risk caused by inadequate storm drainage.

*AC-The following is the text in the reference paper that seems to indicate the statement that we used in the current paper. "Climatic extremities coupled with haphazard human intervention and inadequate planning to handle high storm events led to Uttarakhand flood in July 2013 causing 580 deaths and over 5400 people missing in the aftermath of flood, loss of 9200 cattle and complete damage to 3320 houses (India: Uttarakhand Disaster June 2013). Heavy flooding due to unseasonal rainfall submerged Kashmir twice in a short span of 6 months, September 2014–March 2015, causing over 200 deaths alone in September 2014. Improper drainage system coupled with unchecked and ill-planned urbanization makes the region even more vulnerable to such disasters (The Times of India 2015; The Hindu 2015)." But, as the reviewer has provided more explicit detail on these events, we will update the sentence to reference the Mumbai flood instead of the more recent events.*

*PCiM- the reference flood event will be changed to the Mumbai flood of 2005*

C6R2

Line 165-168: These line should come in the last of introduction section where authors generally list down the objectives or novelty of their work.

*AC-Agree with comment as the authors intended section 2 to part of the overall introduction.*

*PCiM-The section numbers will be adjusted to better reflect that intention*

C7R2

Line 231-232: Cite the NOAA ATLAS like any other technical report and list in the reference. Table 2: Use consistent unit for all the design rainfall.

*AC and PCiM-Agree and will correct in the manuscript.*

C8R2

Line 291: There is no reference for Figure SPM7(a)(IPCC 2014) in reference section. This Figure can be adopted from the source in the manuscript.

*AC and PCiM- A reference will be added as per the reviewer's suggestion.*

C9R2

Figure 1: What do those lines in Orange, magenta, and Black depict? Proper legends discussing each feature must be included with the figure to make it meaningful. The backdrop can be removed as it is making the image complex to understand.

*AC- the Aerial background for the image provides important context the landuse within the catchment. The nodes and links represent the model layout. An explanation of the links and nodes along with the colour difference will be added along with adding the following legend to the figure.*

*PCiM-The following text will be added to the paper in line 220.*
*'The Orange links are example of the sewer network geometry in the model. The blue links represent reaches that are open channel. The magenta links are the surface overflow routes that capture flow that tends to flood in areas and spread outside the sewer network. The black links provides connectivity when the georeferenced locations of nodes are geographically different to the ends of some of the sewer network. The black links provide connectivity in the model.'*

*The figure will be modified as follows;*

C10R2

Figure 6: Figure caption can be shortened as "Comparison of total volume of rainfall and temporal patterns variability impact on peak flood depth. Flood depth variation due to the 6 different temporal patterns with 160 mm of rain compared to 110, 160 and 210 mm of total rainfall over 24 hours distributed over (a) Q1-50 temporal pattern (b) Q3-50 temporal pattern. Flood depths were standardised by subtracting the mean at each location for ease of comparison"

*AC-Agree with the suggested change to Figure 6 caption and we will make the change.*

*PCiM- Caption will be changed to;*
*Comparison of total volume of rainfall and temporal patterns variability impact on peak flood depth. Flood depth variation due to the 6 different temporal patterns with 160 mm of rain compared to 110, 160 and 210 mm of total rainfall over 24 hours distributed over (a) Q1-50 temporal pattern (b) Q3-50 temporal pattern. Flood depths were standardised by subtracting the mean at each location for ease of comparison.*

C11R2

Figure 7: Increase the font size of legends.

*AC and PCiM-Agree with comment and will update the figure appropriately*

*Reference*

*Bisht, D. S., Chatterjee, C., Kalakoti, S., Upadhyay, P., Sahoo, M., and Panda, A.: Modeling urban floods and drainage using SWMM and MIKE URBAN: a case study, Natural Hazards, 84, 749-776, 2016.*

*Peters, G. P., Andrew, R. M., Boden, T., Canadell, J. G., Ciais, P., Le Quere, C., Marland, G., Raupach, M. R., and Wilson, C.: The challenge to keep global warming below 2 [deg]C, Nature Clim. Change, 3, 4-6, 2013.*

---

## Author Response (AR1)

**Responses to Editor**

*We thank the editor for the review of our manuscript and the opportunity to re-submit the revised paper. We agree that the comments have served to make the paper stronger and have addressed all the comments to the best of our ability. We have added description on the model setup within the manuscript as well as provided additional background and detail on the model build information in the supplementary information document . A figure and narrative have been added to help explain the temporal pattern terminology and descriptions as well as further discussion on selecting the scaling methodology and the RCP8.5 scenario. All the Figures have been revised to address the comments of the reviewers. Please note that all text that have been substantially revised, edited, or added is indicated in blue to help track the changes made to the manuscript. These revisions address the reviewers' and editor's comments on streamlining the abstract and introduction as well as improved discussion of the results presented. We have also revised the title of the paper as suggested by the editor. Along with the background information on the model setup, the supplementary information includes results of an additional analysis done in response to a reviewer comment and depth results that are associated with the main manuscript. Please note that the detailed responses to both reviewers ( included below) have been updated to reflect the changes made to the revised manuscript and the updated line numbers where the changes were made.*

*Editor Comments to the Author:*
I have now received reports from two referees and thank the authors for their detailed responses to the referees' comments (a third referee did not provide their report on time). Based on these two reviews and my own reading of the manuscript, I find that the paper fits the aims and scope of HESS, but that the manuscript requires some improvements.
Improvements can be made specifically by: providing further methodological details (on the model setup; the rainfall temporal patterns); providing discussion of the methodological choices (including rationale for choosing the RCP8.5 scenario; discussion of chosen scaling methods); and improving the figures (including font size of legends where reasonable, incl. the revised Figure 1). Additionally, I find that (1) the title should reflect the fact that the study is based on one watershed; (2) the abstract and introduction could be streamlined for greater clarity (e.g. in the abstract, it might be clearer to describe the research gap before introducing the research questions; in the introduction, some of the sentences feel a little disconnected, e.g. adaptation is not defined); (3) sections 4.2-5 are mostly descriptive, and could provide more insight on the implications of the work and how these findings relate to the previous literature.
Both referees have requested to review the paper again after corrections have been made, and I consider that to be a fair request. I would therefore like to invite the authors to upload a revised manuscript for further review by the referees, incorporating the proposed changes and additions, and making any other modifications where they see fit. I look forward to receiving the revised manuscript.

**Response to Reviewer 1**

This manuscript addresses the impact of the changes in temporal pattern and volume of rainfall due to climate change on urban floods. In addition to the impact of total change in rainfall, the impact of the projected changes in temporal patterns alone is estimated. The background scientific question is important and the results are interesting. However, there are several issues that should be addressed before it is published.

*We thank the reviewer for their time and positive assessment of the manuscript. We address the reviewer's concerns in turn with our responses in italics. Please note that the author comment (AC) and Changes in Manuscript (CiM) based on the comments are indicated as such separately for each comment.*

**Major comments**

MC1R1

My major concern is the applicability of the scaling methods (both for volume and temporal pattern) for estimating the "projected" changes in the rainfall. The scaling factors are based on the relationship between the rainfall and temperature in the present climate. However, the present manuscript uses the scaling factor to estimate the "projected" changes in the rainfall induced by the climate change. Both the temporal pattern and rainfall volume will be affected by the changes in various dynamic and thermodynamic factors, not only by the changes in temperature. The applicability of the scaling method, which is based on the present climate variability, to the estimation of changes in rainfall under climate change should be verified. At least, it should be discussed in the manuscript.

*AC- In this work we assume that temperature is the primary climatic variable associated with changing rainfall and have adopted a scaling of 4.7% per degree Celsius. This value appears to be consistent both with historical trends and climate change projections.*
*Figure 4c of Barbero et al (2017) looks at a non-stationary extreme value analysis and finds a sensitivity of approximately 7%/ºC for a non-stationary Theil-Sen estimator for North America. Globally, Westra et al., (2013) find historical trends have global sensitivity between 5.9%/ºC and 7.7%/ºC. However, Kharin et al (2013) report an approximately 4% sensitivity over land globally from the CMIP5 model results with a range of 2.5-5% for the U.S.A.*
*In regard to the evidence above we believe our projections are consistent with the available evidence regarding precipitation change. There is a possibility that it slightly underestimates historical trends, but is at the upper end of predictions from climate model predictions.*
*Finally, there are a number of published works which show that temperature is a recommended covariate for projecting rainfall e.g. Agilan and Umamahesh (2017) and*

*Ali and Mishra (2017) and may indeed implicitly account for dynamic factors (Wasko and Sharma (2017).*

*CiM- The above discussion on sensitivity was added at line 376 to help the reader evaluate the scaling used. We note that the value of 2.92 at Line 370 is a typo and indeed should be 4.7% (as shown correctly in Figure 4).*

*A discussion regarding the validity of using temperature as a covariate for projecting rainfall would be included in the modified manuscript at line 307 to expand on the justification of using temperature scaling beyond the reference to Lenderink and Attema (2015).*

MC2R1
L231-232: The characteristics of the temporal pattern selected from NOAA ATLAS is important in this study. It should be explained more in the manuscript or figures about what the six temporal patterns are like.

*AC- We agree with this comment. The quartiles indicate the timing of the greatest percentage of total rainfall that occurs during a storm. First quartile would indicate that the majority of the rainfall including the peak will occur in the 1st ¼ of the duration, which is between hours 1 through 6 in the case of a 24-hour storm (As indicated in chart a). The distributions were further analysed to determine the frequency of occurrence within each quartile to determine a percentile for each distribution.*

*CiM-Figure R1 was added to the manuscript, which shows the different patterns that, were used in this manuscript. Further, the following text was added at line 286-'The quartiles indicate the timing of the greatest percentage of total rainfall that occurs during a storm. First quartile would indicate that the majority of the rainfall including the peak would occur in the 1st ¼ of the duration, which is between hours 1 through 6 in the case of a 24-hour storm. The temporal distributions were also separated in Atlas 14 to determine the frequency of occurrence within each quartile to determine a percentile for each distribution."*
*.*

[Figure]

*Figure R1. NOAA Atlas 14 temporal patterns used in the modelling*

**Reviewer 1 Minor Comments**

C1R1
L224-226: How was the spatial distribution of rainfall in the catchment considered? Is it uniform over the catchment? Please describe it in the manuscript.

*AC- Yes, the rainfall is assumed to be uniform over the catchment.*
*CiM- A statement on the spatial distribution of rainfall was added in the manuscript at Line 260.*

C2R1
Table 2 (Design Rainfall): Why don't you use the same unit (e.g., mm/24hour) for all three rainfalls?

*AC- We agree and thank you for catching that oversight. We will correct the table to ensure all rainfalls appear in mm.*
*CiM- The table was updated to;*

**Table 2. Description of notation used in reference to the modelled storm depths and temporal distributions (NOAA Atlas 14 volume 8 appendix 5)**

| Design Rainfall | Description |
|---|---|
| 160 mm  24 hour | 2 % exceedance 24-hour duration (50-year return period) rainfall depth |
| 125 mm 24 hour | Lower margin of the 90% confidence interval of the 2 % exceedance 24-hour duration (50-year return period) rainfall depth-Approximately Equivalent to the 20-year 24 hour ARI |
| 210 mm 24 hour | Upper margin of the 90% confidence interval of the 2 % exceedance 24-hour duration (50-year return period) rainfall depth-Approximately Equivalent to the 200-year 24 hour ARI |

C3R1
Table 2 (descriptions of temporal patterns): I don't understand what the "1st quantile 10th percentile" is. Explaining more about the temporal pattern will help reader's better understanding. To show the shape of the pattern in the figure may be helpful.

*AC-As explained in the response above, the quartiles indicate the timing of the greatest percentage of total rainfall that occurs during a storm.  First quartile would indicate that the majority of the rainfall including the peak will occur in the 1st ¼ of the duration, which is between hours 1 through 6 in the case of a 24-hour storm.  The percentile indicates the frequency of occurrence of each pattern within each quartile. In general, the percentile indicates the level of intensity within each quartile with a lower percentile referring to a higher intensity and a lower probability of occurrence. The review is right that this was not adequately explained in the original manuscript and this discussion will be added to the paper.*

*CiM- Please refer to response in comment MC2R1*

C4R1

L264-270: Using some equations for the explanation on the volume scaling may be helpful for readers.

*AC-Reviewer 2 also commented on the relatively short explanation of the methodology. The text at lines 314-328 will be expanded as per below:*

*CiM- Lines 314-328 was expanded to include;*
*"Using established methods (Hardwick Jones et al., 2010a; Utsumi et al., 2011; Wasko and*

*Sharma, 2014), the volume scaling for the 24 hour storm duration was calculated using an exponential regression. The results are presented in Figure 4. First, daily rainfall was paired with daily average temperature. The rainfall-temperature pairs were binned on 2°C temperature bins, overlapping with steps of one degree. For each 2°C bin a Generalized Pareto Distribution fitted to the rainfall data in the bin that was above the 99th percentile to find extreme rainfall percentiles (Lenderink et al., 2011; Lenderink and van Meijgaard, 2008). Extreme percentiles below the 99th percentile (inclusive) were calculated empirically. A linear regression was subsequently fitted to the fitted log-transformed extreme percentiles and used as the rainfall volume scaling (Figure 4). Hence the volume (V) is related to a change in temperature (T) by*

$$V_2 = V_1(1 + \alpha)^{\Delta T}$$

*Where α is the scaling of the precipitation per degree change in temperature."*

C5R1

L399 ".. as shown in Figure 5(a)": Should be Figure 6(a)?

*AC- The Figure reference in the manuscript is correct as is. The intent is to show that the results at Location A are similar.*

*CiM- This section has been revised to reflect the comments and the updated Figure 7*

C6R1

L443-445 "...the mean of the flood depth for projected events does exceed the upper limit of the variability in flood depths for the base scenario": I don't know which part of the figure shows the upper limit of the variability for the base scenario.

*AC-We agree that this statement was a bit vague.*
*CiM- The discussion has been re-written based on the comments from line 524 - 539."*

C7R1

L477-478 "The increase .. due to changes to temporal patterns alone range from 1% to 35%": Does these percentage numbers come from Figure 7? Since the unit of the Figure 7 is meter, it is difficult to figure out the percentage change from Figure 7.

*AC-The percentages are based on the results that were used to generate Figures 7, 8 and 8. Tables showing percentage calculations will be added to the supplemental information to make these results more clear.*

*CiM- The following tables was added in Supplementary Information*

**Impact on flood depth from projected temporal pattern**

|        | A   | B   | C   | E   | D   | F   | G   |
|--------|-----|-----|-----|-----|-----|-----|-----|
| Q1-10  | 1%  | 1%  | 1%  | 0%  | 2%  | 3%  | 0%  |
| Q1-50  | 3%  | 3%  | 20% | 8%  | 8%  | 1%  | 7%  |
| Q1-90  | 0%  | 0%  | 12% | 10% | 13% | 1%  | -1% |
| Q3-10  | 4%  | 5%  | 19% | 10% | 16% | 2%  | 9%  |
| Q3-50  | 5%  | 8%  | 18% | 10% | 35% | 2%  | 9%  |
| Q3-90  | 4%  | 5%  | 16% | 7%  | 12% | 15% | 1%  |

**Impact on flood depth from projected volume and temporal pattern**

|        | A    | B   | C   | D    | E   | F    | G   |
|--------|------|-----|-----|------|-----|------|-----|
| Q1-10  | 13%  | 40% | 53% | 95%  | 21% | 108% | 12% |
| Q1-50  | 31%  | 51% | 74% | 69%  | 37% | 57%  | 25% |
| Q1-90  | 17%  | 34% | 49% | 37%  | 27% | 9%   | 35% |
| Q3-10  | 27%  | 46% | 61% | 147% | 59% | 76%  | 22% |
| Q3-50  | 26%  | 52% | 68% | 173% | 57% | 119% | 17% |
| Q3-90  | 23%  | 48% | 78% | 140% | 49% | 170% | 8%  |

*References:*
*Ali, H., and V. Mishra (2017), Contrasting response of rainfall extremes to increase in surface air and dewpoint temperatures at urban locations in India, Sci. Rep., 7(1), 1228, doi:10.1038/s41598-017-01306-1.*
*Agilan, V., and N. V Umamahesh (2017), What are the best covariates for developing non-stationary rainfall Intensity-Duration-Frequency relationship?, Adv. Water Resour., 101, 11–22, doi:10.1016/j.advwatres.2016.12.016.*
*Barbero, R., H. J. Fowler, G. Lenderink, and S. Blenkinsop (2017), Is the intensification of precipitation extremes with global warming better detected at hourly than daily resolutions?, Geophys. Res. Lett., 1–10, doi:10.1002/2016GL071917.*
*Kharin, V. V., F. W. Zwiers, X. Zhang, and M. Wehner (2013), Changes in temperature and precipitation extremes in the CMIP5 ensemble, Clim. Change, 119(2), 345–357, doi:10.1007/s10584-013-0705-8.*
*Westra, S., L. Alexander, and F. Zwiers (2013), Global increasing trends in annual maximum daily precipitation, J. Clim., 26, 3904–3918, doi:http://dx.doi.org/10.1175/JCLI-D-12-00502.1.*

**Responses to Reviewer 2**

This manuscript entitled "Increase in urban flood risk resulting from climate change – The role of storm temporal patterns" draws readers' attention towards importance of storm temporal pattern in urban flood modelling under altering climatic scenario. Given the frequent reporting of urban floods across the globe this study provides useful insight to urban flood modellers. The manuscript fits the aim and scope of HESS quite well, and can be accepted provided authors address the following comments with required modifications and justifiable responses.

*We thank the reviewer for their time. We also thank the reviewer for their favourable assessment of the manuscript content and results presented. We address the reviewers concerns in turn with our responses in italics. Please note that the author comment (AC) and Changes in Manuscript (CiM) based on the comments are indicated as such separately for each comment.*

**Major comments**
MC1R2
Why did authors choose a 50-year return period storm? Why not 10, 20, or 25 years return period that is much common for urban flood modelling studies? Or why not 100 year return period?

*AC- The model was initially set up in the year 2000 and has been continuously updated and maintained over the 15 years since then, based on the prevailing 100-year 24-hour rainfall event for that catchment (100-year selected to set base flood elevation for the open water areas within the catchment). Atlas 14 V8 updated the rainfall and increased the new 100-year rainfall depths such that previous 100-yr rainfall depth is now the 50-year rainfall depth. This implies that the hydraulic constraints pertaining to rainfall depth substantially higher than the original model build value ( which can results from temperature scaling of the revised Atlas 14 100-year depth ) are not in place in the model as it currently stands. Given the work involved in re-specifying these hydraulic constraints to simulate the scaling effects of the new 100-year depth, it is not feasible to do so in the current study. While it is possible to simulate the impact of lower return periods (10, 20 or 25 years), the decision was made that this does not add to the discussion or lead to any new conclusions being reached. For simplicity sake, we decided that the most illustrative presentation still remains the 50-year return period case that is presented. A short discussion on this and the ability to generate results for lower period ARI has been included in the supplemental information.*

MC2R2
RCP 8.5 scenario is derived using the most pessimistic assumption and is very unlikely given the ongoing worldwide efforts to curb the carbon emission and green initiatives. Though such studies using RCP 8.5 gives mind boggling figures, these remain very unlikely. A more likely scenario could be RCP 4.5 should have been used along RCP 8.5 to encompass the effects of climatic change. Secondly, authors carried out the study for projected period for 2081-2100 skipping the intermediate time frames. Is there no significant results during 2025-2050 or 2051-2080? Though the results would be much pronounce in the later part of the century, intermediate time frame should also be discussed. Authors must explain the rationale behind selecting worst case climatic scenario i.e., RCP 8.5 and also come up with the reasoning to skip RCP 4.5 and selection of specific time frames for such modelling exercises for potential users. Additional details related to exercise can be provided in supplementary materials.

*AC-As suggested, the authors did an additional analysis for the case of RCP 4.5, looking at a temperature change of $3^0C$ . The following table illustrates that the trend in results are similar to the RCP 8.5 scenario for all cases as expected. Understandably, the impact to flood depths is not as significant as when looking at a $5^0C$ increase in temperature. The main goal of the paper is to demonstrate the importance of accounting for the changes expected in temporal patterns of rainfall, which looks at relative impacts.*

*We wish to note that there is literature suggesting that we are tracking on a RCP8.5 scenario (Peter. G et al 2013) and indeed many forecasts suggest greater temperature increases over land. The authors agree that there should be rigorous thought on how far out and what level of climate impacts should be considered when selecting a threshold for design or when setting absolute flood depths.*

*CiM- The following table was added in Supplementary Information with explanatory narrative.*

*Table S1- Results showing normalized flood depths around the mean at each location for a projected temperature increase of 3 deg C.*

| | Current (normalized) | | | | |
| --- | --- | --- | --- | --- | --- |
| | A | B | D | C | G |
| Q1-10 | 0.58 | 0.15 | 0.65 | 0.53 | 0.33 |
| Q1-50 | -0.28 | -0.07 | -0.08 | -0.11 | -0.16 |
| Q1-90 | -0.31 | -0.10 | -0.23 | -0.24 | -0.37 |
| Q3-10 | -0.14 | -0.04 | -0.19 | -0.12 | -0.07 |
| Q3-50 | -0.03 | -0.01 | -0.18 | -0.08 | 0.03 |
| Q3-90 | 0.18 | 0.07 | 0.03 | 0.01 | 0.24 |

| | Projected patterns (normalized) | | | | |
| --- | --- | --- | --- | --- | --- |
| | A | B | D | C | G |
| Q1-10 | 0.60 | 0.16 | 0.67 | 0.54 | 0.33 |
| Q1-50 | -0.21 | -0.06 | -0.05 | -0.04 | -0.09 |
| Q1-90 | -0.31 | -0.10 | -0.19 | -0.21 | -0.38 |
| Q3-10 | -0.05 | -0.02 | -0.15 | -0.06 | 0.03 |
| Q3-50 | 0.11 | 0.02 | -0.07 | -0.02 | 0.16 |
| Q3-90 | 0.27 | 0.10 | 0.08 | 0.06 | 0.25 |
| *Change in Mean* | *0.07* | *0.02* | *0.05* | *0.04* | *0.05* |

| | Projected patterns and volumes (normalized) | | | | |
|---|---|---|---|---|---|
| | A | B | D | C | G |
| Q1-10 | 0.98 | 0.38 | 1.36 | 0.73 | 0.49 |
| Q1-50 | 0.34 | 0.14 | 0.04 | 0.09 | 0.25 |
| Q1-90 | 0.04 | 0.03 | -0.15 | -0.15 | -0.09 |
| Q3-10 | 0.47 | 0.16 | 0.15 | 0.03 | 0.27 |
| Q3-50 | 0.91 | 0.36 | 0.60 | 0.23 | 0.34 |
| Q3-90 | 0.91 | 0.36 | 0.60 | 0.23 | 0.34 |
| *Change in Mean* | *0.61* | *0.24* | *0.44* | *0.19* | *0.27* |

MC3R2
The study employs the modelling component in a big way to derive the conclusions, however, there is no discussion made on how the modelling framework was setup. Catchment sizes in the modelling setup varies from 0.25 sq km to 22 sq km that makes almost 90 times change in smallest and largest catchment. Interestingly, unlike river basin scale studies in urban drainage modelling catchment boundaries are not demarcated by their natural topography as the interceptor drains divert the runoff water omitting the natural stream lines. How the authors have discretised such vastly different sized catchments? Authors should discuss how the impervious area is estimated to include in modelling framework, and other parameters used in the modelling exercise should be tabulated. Did authors fed the existing storm sewage network into the model to rout the flow from a particular sub-catchment to outlet or directed them directly to the outlet from the subcatchment? Also discuss how the model was calibrated and validated. A separate section on model setup is highly warranted to make the manuscript more informative.

*AC- As mentioned in the response above, the model used in the study was set initially in year 2000 and has been continuously maintained and updated to include the latest available landuse/landcover and stormwater infrastructure information. The model includes all components of stormwater conveyance within the catchment including sewers, open channels and storage areas, along with street overflows. Highly detailed delineation of both sub-catchment boundaries and impervious area was done using a high resolution DEM, development construction and grading plan overlays and aerial imagery within a GIS environment. All surface runoff is fed into the appropriate inflow points of the hydraulic conveyance system. The model has been validated and used to design major capital improvement and flood mitigation projects over the years. The following link connects to a report that discusses extensive model validation work based on an extreme storm.*

*http://www.swwdmn.org/pdf/projects/completed/2006%20Stormwater%20Modelin g%20Report_HDR.pdf*
*CiM – Additional detail on the construction and background information was added in lines 188-214. A more detailed discussion and example of background data was added in Section 1 of the Supplementary information. The following references was be added in the Supplementary information.*

*Model Development and related information:*
*Hettiaracchchi, S, and W. Johnson (2006), Stormwater modelling Report, HDR Project No. 32072, [available online:*
*http://www.swwdmn.org/pdf/projects/completed/2006%20Stormwater%20Modelin g%20Report_HDR.pdf]*

*The following is an additional presentation that discusses flood mitigation projects analysed using this model.*

*Hettiarachchi. S, Beduhn. R, Christopherson. J Moore. M, Managing Surface Water for Flood Damage Reductio, World Water and Environmental Resources Congress 2005, May 15-19, 2005 | Anchorage, Alaska, United States, doi-10.1061/40792(173)321*

*Many of the projects listed here are based on using this model.*
*http://www.swwdmn.org/projects/*

MC4R2
Line 266-267 and Figure 4: "The rainfall-temperature pairs were binned on 2 degree temperature bins . . ." Does it mean that binning was done by counting the number of rainfall events and their corresponding magnitudes at each 2 degree temperature interval? What does the height of each bin depict? What do the count and precipitation magnitudes from primary and secondary y-axis show?

*AC-The first reviewer also commented on the clarity of this paragraph. We believe some confusion arose from the histogram in Figure 4 and having two sets of axes. The histogram would have effectively only shown every second bin (as the binning is performed using two degree bins with overlapping steps of one degree).*
*CiM- Lines 314-328 was expanded to read as follows and Figure 5 was replaced with the figure below:"Using established methods (Hardwick Jones et al., 2010a; Utsumi et al., 2011; Wasko and*
*Sharma, 2014), the volume scaling for the 24 hour storm duration was calculated using an exponential regression. The results are presented in Figure 4. First, daily rainfall was paired with daily average temperature. The rainfall-temperature pairs were binned on 2°C temperature bins, overlapping with steps of one degree. For each 2°C bin a Generalized Pareto Distribution fitted to the rainfall data in the bin that was above the 99th percentile to find extreme rainfall percentiles (Lenderink et al., 2011; Lenderink and van Meijgaard, 2008). Extreme percentiles below the 99th percentile (inclusive) were calculated empirically. A linear regression was subsequently fitted to*

*the fitted log-transformed extreme percentiles and used as the rainfall volume scaling (Figure 4). Hence the volume (V) is related to a change in temperature (T) by*

$$V_2 = V_1(1 + \alpha)^{\Delta T}$$

*Where $\alpha$ is the scaling of the precipitation per degree change in temperature."*

*In light of the above comments Figure 4 has been modified and the new figure is shown below. The histogram in the figure has been removed to prevent confusion as the fitted quantiles were not necessarily matching the histogram bins presented creating ambiguity in the results.*

[Figure]

*Figure 5 Scaling total volume of rainfall with temperature for Minneapolis (1901-2014 daily rainfall). Grey dots are rainfall temperature-pairs and the coloured dots are the extreme percentiles. The grey dashed line represents a scaling of 7 %.*

MC5R2

In Line 339-340 authors say "The flood depths extracted from the model were first analysed to compare variability between temporal patterns and total rainfall depth. . ." SWMM is a 1-dimensional model and does not simulate the flood extent or flood depth. Though it simulates depth of water being flooded from a node, it depends on the adequacy of drainage network. While discussing the flood depth in relation to urban scenario, the depth of flood inundation should be used rather that the depth of total water flooded from a particular node or from the entire system. This aspect need some clarification.

*AC- The review is correct in how SWWM typically shows flooded nodes and yes, the flood depth are dependent on the adequacy of the model. As discussed in the comment regarding the background and extent of the model build, extensive surface overflow routes from flooded areas as well as explicitly modelling street overflows and storage extents are included in the model geometry. This allows the resulting flood depths to take into account the flood extents as well as opposed to the typical funnel that is SWMM uses at nodes. Additionally, majority of the reference locations are at local storage nodes that provide a good representation of flood extents. Storage nodes have depth/area curves that represent flood extents at each depth. Therefore, the results from this model provide a reasonably accurate representation of extents related to each flood depth.*
*CiM- This discussion was added to model discussion at line 188 and in the supplementary information.*

MC6R2

In Line 342-343 authors say "These sub-models show the variation in catchment response
to runoff generated by different land use types. . ." There is no provision of feeding LULC information in SWMM, rather it takes percent pervious and impervious area. Different land use types gives a notion that model is simulating overland flow explicitly for residential, paved surfaces, parks, grassed land etc. How the different land use land cover type were incorporated in the model? Similarly, in Line 360 and 368 authors talk about "local storage/ local natural storage". How these storage was incorporated into the modelling exercise.

*AC- Agree with the comment that SWMM does not have provisions to explicitly designate LULC in a runoff area. However, as discussed in response to comment 3, the model was setup in extensive detail using multiple layers of information that provide characteristic percentages of impervious area based on the built environment within each of the sub-catchments. By discretising to small areas, the model then is able to isolate the various landuse types within each catchment and generate a composite impervious percentage and a rate of runoff representative of each different landuse type. Local constructed storage refers to stormwater ponds that were built as part of rate and volume controls to meet post development rules requirements. Natural storage locations refer to existing ponds and lakes within the catchments. The storage information is added into the model as depth/area tables using the DEM and*

*bathymetric survey (major storage locations) for natural storage locations and construction plans for constructed storage locations.*
*CiM- N/A*

MC7R2
Line 399-411 does not helps much and as a reader I find it less convincing how Fig 6(a) is different than Fig. 6(b) and how pronounce the difference is for temporal pattern case and total rainfall volume case. Moreover, visually Figure 6(a) and (b) are seems more or less identical with little change. It would be better if author can redraw them to convey their point. Perhaps, comparison of Q1-50 and Q3-50 in same graph for temporal pattern variation or total rain volume variation will help the readers' understanding. Also specific markers for different cases should be provided, as of now there are 4 squares and each belong to which requires a thorough reading. Make the image self-explanatory.

*AC- We agree with the reviewer and we will redraw Figure 6 to convey the intended point. Figure 6 (a) and (b) attempts to illustrate the variation between temporal pattern vs volume of rainfall, and is not intended to show changes based on a particular, or each temporal pattern. The fact that Figure 6a and 6b are similar shows that this variability is generally independent of the temporal pattern chosen for the volume variability. That is to say, that the results are not skewed to be favourable by picking a single temporal pattern to examine the volume variability.*
*CiM- Figure 6 (a) and (b) was modified (Figure 7) to emphasize the range of results as well as the comparison between current and projected results.*

*MC8R2*
Fourth conclusion suggests the 'increase in potential flood risk purely due changes to "how it rains" as a result of climate change impacts'. This conclusion is drawn from the analysis shown in Fig 6 and Fig. 7. How the temporal pattern variation has a pronounce effect on flood risk as from the Fig. 6 gives almost same picture for temporal pattern for Q1 and Q3 rainfall, whereas from Fig. 7 also not much significant change can be noticed in the standardized flood depth due to current temporal patterns and projected patterns unlike Fig. 8, where the difference is really remarkable. An elaboration would help the readers' understanding.

*AC-The reviewer is correct that the 4th conclusion is based on the data that was used to generate Figure 7. This conclusion points to flood impacts that are due to projected temporal patterns. Whereas Figure 8 shows flood impacts due to projection of both temporal patterns and rainfall volumes.*
*CiM-The discussion was revised in response to the comments and the results tables were added to the supplementary information.*

MC9R2
Temporal pattern or distribution used from NOAA ATLAS should be discussed in short. It's not clear what does nth quartile at mth percentile means. It would be insightful if authors show it in figure.

*AC- We agree with this comment as well as the comment by the other reviewer. Figure R1 will be added to the manuscript which shows the different patterns that were used in this manuscript.*

*CiM-Figure 3 was be added to the manuscript which shows the different patterns that were used in this manuscript. Further, the following text was added at line 257-'The quartiles indicate the timing of the greatest percentage of total rainfall that occurs during a storm. First quartile would indicate that the majority of the rainfall including the peak will occur in the 1st ¼ of the duration, which is between hours 1 through 6 in the case of a 24-hour storm. The temporal distributions were also separated in Atlas 14 to determine the frequency of occurrence within each quartile to determine a percentile for each distribution."*

.

[Figure]

*Figure R1. NOAA Atlas 14 temporal patterns used in the modelling*

MC10R2
In Line 283, what does author mean by "current industry standard temporal distributions"?: Authors may like to use supplementary material space for elaborate discussion to clarify the doubt.

*AC- Temporal patterns from design guidelines and standards are used throughout civil engineering and consulting industry for design flood estimation and these standards are commonly referred as 'industry standards'.*

*CiM- To clarify what is meant by this the text at line 339 is replaced with "temporal patterns for design flood estimation" instead of "current industry standard" and refer to the example of the NOAA Atlas 14 temporal patterns.*

**minor comments**

C1R2

First line of abstract [Line 8-9] i.e., "Warming temp . . ." is almost repeated in [Line 18-19] i.e., "Current literature . . ."

*AC- Agree and will remove the first sentence to prevent repetition.*

*CiM- Abstract has been revised to reflect the comments*

C2R2

Fix the citation formats throughout the text, for example in [Line 89] the citation should be like Milly et al. (2007).

*AC-Agreed and appreciate noting the need to adjust the citation.*

*CiM- The citations were edited appropriately so that the parenthesis occur in the correct location.*

C3R2

Delete 'an' before EPA-SWMM in [Line 182], delete '2016' after EPA in [Line 185]

*AC and PCiM- Thank you. This edit was done at line 182*

C4R2

Line 64: Correct the "Intensity/Duration/Frequency" as "Intensity-Duration-Frequency"

*AC and CiM-Was update in the  paper to reflect suggested change*

C5R2

Line 114-116: It would not be apt to link Uttarakhand and Kashmir floods in India with poor storm sewer design from Bisht et al. (2016). As these floods were caused by cloud burst and moreover the topography is hilly in that place. However, Bisht et al. (2016) discussed the Mumbai flood that can be aptly link with flood risk caused by inadequate storm drainage.

*AC-The following is the text in the reference paper that seems to indicate the statement that we used in the current paper. "Climatic extremities coupled with haphazard human intervention and inadequate planning to handle high storm events led to Uttarakhand flood in July 2013 causing 580 deaths and over 5400 people missing in the aftermath of flood, loss of 9200 cattle and complete damage to 3320 houses (India: Uttarakhand Disaster June 2013). Heavy flooding due to unseasonal rainfall submerged Kashmir twice in a short span of 6 months, September 2014–March 2015, causing over 200 deaths alone in September 2014. Improper drainage system coupled with unchecked and ill-planned urbanization makes the region even more vulnerable to such disasters (The Times of India 2015; The Hindu 2015)." But, as the reviewer has provided more explicit detail on these events, we will update the sentence to reference the Mumbai flood instead of the more recent events.*

*CiM- the reference flood event was changed to the Mumbai flood of 2005*

C6R2

Line 165-168: These line should come in the last of introduction section where authors generally list down the objectives or novelty of their work.

*AC-Agree with comment as the authors intended section 2 to part of the overall introduction.*

*CiM-The section numbers and the write up was revised to better reflect that intention*

C7R2

Line 231-232: Cite the NOAA ATLAS like any other technical report and list in the reference. Table 2: Use consistent unit for all the design rainfall.

*AC and CiM-Agree and was correct in the manuscript.*

C8R2

Line 291: There is no reference for Figure SPM7(a)(IPCC 2014) in reference section. This Figure can be adopted from the source in the manuscript.

*AC and PCiM- A reference was added as per the reviewer's suggestion.*

C9R2

Figure 1: What do those lines in Orange, magenta, and Black depict? Proper legends discussing each feature must be included with the figure to make it meaningful. The backdrop can be removed as it is making the image complex to understand.

*AC- the Aerial background for the image provides important context the landuse within the catchment. The nodes and links represent the model layout. An explanation of the links and nodes along with the colour difference will be added along with adding the following legend to the figure.*

*CiM-The following text was added to the paper in line 227.*
*'The Orange links are example of the sewer network geometry in the model. The blue links represent reaches that are open channel. The magenta links are the surface overflow routes that capture flow that tends to flood in areas and spread outside the sewer network. The black links provides connectivity when the georeferenced locations of nodes are geographically different to the ends of some of the sewer network. The black links provide connectivity in the model.'*

*The figure was modified as follows;*

C10R2

Figure 6: Figure caption can be shortened as "Comparison of total volume of rainfall and temporal patterns variability impact on peak flood depth. Flood depth variation due to the 6 different temporal patterns with 160 mm of rain compared to 110, 160 and 210 mm of total rainfall over 24 hours distributed over (a) Q1-50 temporal pattern (b) Q3-50 temporal pattern. Flood depths were standardised by subtracting the mean at each location for ease of comparison"

*AC-Agree with the suggested change to Figure 6 caption and we will make the change.*

*CiM- Figure and caption revised based on comments and is now Figure 7*

C11R2

Figure 7: Increase the font size of legends.

*AC and CiM-Agree with comment and update the figure appropriately as Figures 8 and 9*

*Reference*

[revised manuscript text omitted]

---

## Author Response (AR2)

**Response to Editor**

*We thank the editor for working with the referees on our revised paper and are excited to submit the revised final version. We agree with the minor comments and fixes suggested and have addressed them to the best of our ability. Following are the responses to each of the comments made by the reviewers. Please note that we did not include the updated figures in this response document as most of them were minor fixes and have been updated in the revised marked-up manuscript attached.*

**Editors Comments to Authors**

Thank you for your detailed and thorough revisions to the manuscript, which has now been re-reviewed by the two referees. They have both made some minor suggestions for improvements/clarification, which seem reasonable and should be straightforward to address. I would therefore like to invite you to submit a revised final version, taking these suggestions into account.

**Responses to Reviewer 1**

Most of my last comments are adequately addressed in the revision. However, I still have some minor comments. I think some parts lack information that is necessary for readers.

*We thank the reviewer for their time and effort to re-review the revised manuscript. We agree with the minor comments and address the reviewer's comments and suggested changes in turn with our responses in italics. Please note that Changes in Manuscript (CiM) based on the comments are indicated as such separately for each comment.*

**Minor comments.**

Comment 1
- L260: "The spatial distribution of rainfall was kept constant for this study". It gives me an impression that the precipitation is not uniform over the region and the spatial distribution does not change with global warming. If the precipitation is uniform over the region, please make it clear.

*CiM- Line 260 was edited as " The spatial distribution of rainfall is assumed to be uniform for this study"*

Comment 2
- Figure 3. Does the "N-th fraction" mean the fraction with N-th rank or just the N-th four-hour period (e.g., 1st=0:00-4:00, 2nd=4:00-8:00, ...) in the 24-hour? If it is the former one (I believe so, from the main text), it would be helpful for readers if they are called "N-th ranked fraction".

*CiM- Figure 3 was updated to reflect the appropriate Fraction numbers and the line " F# represents the # ranked fraction" was added at line 391*

Comment 3
- L416-418: I cannot find where the number "30% (1.3m) variation in flood depth (relative to mean flood depth)" comes from in Figure 6(a). Also, I cannot see what is "mean flood depth". Please clarify them.

*CiM – We agree that the percentage creates some confusion and the "30%" reference was deleted as the more significant impact is the 1.3 m variation in flood depth. Line 415 was updated as "A striking result is the approximately 1.3 m variation in flood depth at Wilmes Lake purely due to variation of how the rain falls within the duration of the storm".*

Comment 4
- L426-427: Again, I cannot find where the number "9.5%" comes from. Are you comparing the area under each line in Fig.6(a)? Please clarify it.

*CiM- As with the previous comment, we agree that this percentage reference needs clarification and the calculations for this value was added to the supplemental information.*

Comment 5
- Figure 8 and Figure 9: What are the solid squares? Please describe it in the figures or captions.

*CiM – The legend in the Figure was update to reference the mean depth.*

Comment 6
- L548:"The mean flood depth". Please show the mean flood depth in the figures (Fig.9) (do the solid squares show the means?).

*CiM – Yes, the solid squares show the mean flood depth which has been added to the legend in Figures 8 and 9.*

Comment 7
- L600: consider --> considered.

*CiM- We thank the reviewer for catching this typo which was corrected as mentioned in the comment.*

**Responses to Reviewer 2**

I am satisfied with the responses and modifications made by the authors. I have few suggestions for minor fixes that I have listed below.

*We thank the reviewer for their time and effort to re-review the revised manuscript. We address the suggested minor fixes in turn with our responses in italics. Please note that Changes in Manuscript (CiM) based on the comments are indicated as such separately for each comment*

**Minor Fixes:**
Comment 1
Fig. 1: The scale division should be corrected. Further there is "Violet" line in legend that is not visible in Figure. Yellow circle represent nodes should be mentioned. There are 2 scales in Figure, one in "km" another in "miles", please make them consistence.

*CiM- Agree with the comment that the scale shown in the insert is not required. The Figure has been updated along with the legend to show appropriate symbology. The colours of the lines in the legend were adjusted to try to make them match.*

Comment 2
Fig. 3: I find it difficult to read Y-axis tick marks. There are too many tick marks. Perhaps it can be improved.

*CiM- The Figure was updated to reduce the tick marks as suggested.*

Comment 3
Fig. 4: Give appropriate X-label and Y-label.

*CiM- Figure 4 was updated as suggested by adding a Y- label.*

Comment 4
Fig. 6: Write (a) and (b) inside may be on top right or top middle or top left corner of figures instead writing them with Y-labels.

*CiM Figure 6 was updated by adding "a" and "b" at the top left corner of the figure.*

Comment 5
Fig. 6(a): X-label should be "Time (h)"

*CiM- Figure 6 X label was updated as suggested.*

Comments 6
Fig. 7: Add X-label, and delete "(a)" from the Y-label.

*CiM- The Y label for Figure 7 was updated by deleting "(a)"*

Comments 7
Table 2. It would help if author can include the link as (Accessed from
http://www.nws.noaa.gov/oh/hdsc/PF_documents)

*CiM – The website to the NOAA atlas 14 Vol 8 was added at line 276*

Marked up version of the revised final manuscript with text edits high-lighted in light blue.

[revised manuscript text omitted]